

# Pan-Arctic Aerosol Number Size Distributions: Seasonality and Transport Patterns

Eyal Freud[1], Radovan Krejci[1], Peter Tunved[1], Richard Leaitch[2], Quynh T. Nguyen[3], Andreas Massling[4], Henrik Skov[4], Leonard Barrie[5]

[1]Department of Environmental Science and Analytical Chemistry & Bolin Centre of Climate Research, Stockholm University, Stockholm 10691, Sweden
[2]Climate Research Division, Environment and Climate Change Canada
[3]Department of Engineering, Aarhus University, Aarhus 8200, Denmark
[4]Arctic Research Center, Department of Environmental Science, Aarhus University, Roskilde 4000, Denmark
[5]Department of Geological Sciences & Bolin Centre of Climate Research, Stockholm University, Stockholm 10691, Sweden

*Correspondence to*: Eyal Freud (eyal.freud@aces.su.se)

**Abstract.** The Arctic environment has an amplified response to global climatic change. It is sensitive to human activities that mostly take place elsewhere. For this study, a multi-year set of observed aerosol number size distributions in the diameter range of 10 to 500 nm from five sites around the Arctic Ocean (Alert, Villum Research Station - Station Nord, Zeppelin, Tiksi and Barrow) was assembled and analysed.

A cluster analysis of the aerosol number size distributions, revealed four distinct distributions. Together with Lagrangian air parcel back-trajectories, they were used to link the observed aerosol number size distributions with a variety of transport regimes. This analysis yields insight into aerosol dynamics, transport and removal processes, on both an intra- and inter-monthly scales. For instance, the relative occurrence of aerosol number size distributions that indicate new particle formation (NPF) event is near zero during the dark months, and increases gradually to ~40% from spring to summer, and then collapses in autumn. Also, the likelihood of Arctic Haze aerosols is minimal in summer and peaks in April at all sites.

The residence time of accumulation-mode particles in the Arctic troposphere is typically long enough to allow tracking them back to their source regions. Air flow that passes at low altitude over central Siberia and Western Russia is associated with relatively high concentrations of accumulation-mode particles ($N_{acc}$) at all five sites – often above 150 cm$^{-3}$. There are also indications of air descending into the Arctic boundary layer after transport from lower latitudes.

The analysis of the back-trajectories together with the meteorological fields along them indicates that the main driver of the Arctic annual cycle of $N_{acc}$, on the larger scale, is when atmospheric transport covers the source regions for these particles in the 10-day period preceding the observations in the Arctic. The scavenging of these particles by precipitation is shown to be important on a regional scale and it is most active in summer. Cloud processing is an additional factor that enhances the $N_{acc}$ annual cycle.

There are some consistent differences between the sites that are beyond the year-to-year variability. They are the result of differences in the proximity to the aerosol source regions and to the Arctic Ocean sea-ice edge, as well as in the exposure to free tropospheric air and in precipitation patterns – to mention a few. Hence, for most purposes, aerosol observations from a single Arctic site cannot represent the entire Arctic region. Therefore, the results presented here are a powerful observational benchmark for evaluation of detailed climate and air chemistry modelling studies of aerosols throughout the vast Arctic region.





# 1    Introduction

Aerosols affect climate and weather in various ways. For example, they scatter and or absorb solar radiation, reducing surface insolation and altering the atmospheric radiation budget, which is referred to as "the aerosol direct effect" (Yu et al., 2006). They also affect Earth's radiation budget by altering the cloud and precipitation

properties, which is known as "the aerosol indirect effect" (Lohmann and Feichter, 2005). Meteorology is the main driver of the aerosol life cycle, so changes in weather patterns change the aerosol characteristics, which can in turn feedback and affect the meteorology. Improving the understanding of the aerosol-cloud-climate interactions is therefore crucial for reducing the uncertainties in future climatic projections.

The total radiative forcing by anthropogenic aerosol particles during the Anthropocene is highly uncertain, but

according to the Intergovernmental Panel on Climate Change assessment (IPCC), it is most likely negative (Boucher et al., 2013). This means that it has been masking some of the observed global warming due to increased greenhouse gases. The Arctic region is especially sensitive to perturbations of the radiative budget. It has been shown that the temperature increase rate in the Arctic region has been more than twice that of the global average since the 1980s (Cohen et al., 2014), thus highlighting the sensitive nature of this region. A recent modelling study

by Acosta Navarro et al. (2016) states that one of the main causes for the so-called Arctic Amplification is the reduction in aerosol emissions in recent decades in the developed countries surrounding the Arctic, especially in Europe, consistent with a study showing the overall past influence of the Arctic aerosol as one of cooling (Najafi et al., 2015). However, warming is not homogeneous across the Arctic region and throughout the year.

There were reports of dirty snow on a large scale in the high Arctic (>70°N) already in the 19[th] century (Garrett

and Verzella, 2008). More reports on hazy skies, especially in the springtime, were published later. The signature of anthropogenic activity on this Arctic Haze was revealed only in the late 1970s (Flyger et al., 1980; Heidam and Z., 1984; Quinn et al., 2007; Shaw, 1981), which led to the understanding that long-range transport brings the pollutants to the Arctic from distant sources.

Arctic Haze is characterized by increased atmospheric turbidity as the result of higher-than-average concentrations

of accumulation-mode aerosols (Radke and Lyons, 1984; Rahn et al., 1977). It is often seen as distinct dark bands when flying above the Arctic during daylight. It is accompanied by gaseous constituents, and it exhibits a strong annual cycle with a maximum in spring and a minimum in summer (Barrie, 1986). Inefficient removal processes and a rather stable lower troposphere in winter allow the particles to stay airborne for a long time, and to travel great distances.

There is an increasing number of studies using different approaches to identify the source regions of the major Arctic short-lived pollutants and their seasonality (Croft et al., 2016b; Heidam et al., 2004; Hirdman et al., 2010b; Huang et al., 2015a; Liu et al., 2015; Massling et al., 2015; Nguyen et al., 2013; Polissar et al., 2001; Tunved et al., 2013). All of them indicate that periods with high levels of anthropogenic pollutants, are mostly associated with transport from northern Eurasia to the Arctic sites. These support several previous long-range transport

modelling studies of the origin of Arctic haze (Barrie et al., 1989; Christensen, 1997; Heidam et al., 2004) that showed the dominance of Eurasian sources and the strong transport in the lower troposphere. In addition, (Hirdman et al., 2010a) showed that the long-term decreasing trends in black carbon and sulfate aerosol concentrations in the Arctic are dominated by changes in emissions rather than long-term trends in atmospheric transport patterns.





However, aerosols are not only transported to the Arctic, they are also formed in situ via gas to particle conversion processes. Episodes with high concentrations of nucleation-mode aerosols (dry diameter < ~20 nm), following new particle formation (NPF) events, have been documented in various climatic zones both in the boundary layer and in the free atmosphere (Kulmala et al., 2004). In order to produce a large number of new stable molecular

clusters in the atmosphere, some preconditions are required. These include supersaturation of the condensing vapours, such as the oxidation products of dimethyl-sulphide (DMS) and ammonia, which increases the nucleation rate (Kirkby et al., 2011), together with a low condensation sink, i.e. less particle surface for the molecules to condense upon. The most favourable conditions for NPF formation in the Arctic are in the summer months (e.g. Asmi et al., 2016; Croft et al., 2016a; Leaitch et al., 2013; Nguyen et al., 2016; Tunved et al., 2013), where

sulfuric acid plays a key role, but also in the spring new particle formation occurs but now initiated by $HIO_3$ (Sipilä et al., 2016).

Large-scale atmospheric and oceanic phenomena as well as persistent weather patterns might affect the intra-Arctic as well as year-to-year variability of Arctic Haze and NPF. On daily or weekly timescales, however, the aerosol properties are governed by the synoptic and meteorological conditions, which may induce considerable

variations from the mean annual pattern. This highlights the importance of conducting continuous, long-term and high-resolution aerosol measurements at multiple locations in order to characterize the aerosols across the Arctic throughout the year. Until date, the authors are not aware of any previous study that has compared observations of aerosol number size distributions from multiple Arctic sites and linked them to the atmospheric transport patterns and the general meteorology.

The aim of this paper is to present and discuss the differences and the similarities of the aerosol general characteristics – as inferred from their number size distributions, between different sites across the Arctic, as well as exploring their common transport pathways and the main source regions of the precursor gases and accumulation-mode particles. This allows the assessment of the spatial representativeness of the aerosol measurements at each one of the sites on varying timescales and could provide a benchmark for atmospheric

models with resolved aerosol number size distributions.

## 2    Methodology

### 2.1    Measurement sites and Instrumentation

The foundations of this study are observations of aerosol number size distributions from five Arctic locations (Fig. 1). The Zeppelin research station in western Svalbard (78.9° N, 11.9° E) is right below the top of mount Zeppelin

at an elevation of 474 m (all absolute heights in this paper are with respect to the mean sea level). Villum Research Station - Station Nord (VRS) in northeastern Greenland (81.6° N, 16.7° W, 24 m) is 600 km to the west-northwest of Zeppelin. The measurement site at Alert is 700 km to the west-northwest of Station Nord. Alert is at the northernmost tip of the Canadian Arctic (82.5° N, 62.3° W, 210 m) only 800 km from the North Pole. Point Barrow in northern Alaska, US (71.32° N, 156.6° W, 5 m), is the southernmost site and is 3300 km from Zeppelin

on the opposite side of the Arctic Ocean. The observation site at Tiksi (71.6° N, 128.89° W, 35 m) completes the list and represents the Russian sector of the Arctic.

The Zeppelin research station is located ~2 km south of the small community of Ny-Ålesund, but the elevation difference of 474m as well as the prevailing wind patterns inhibit pollution from nearby sources reaching the



measurement site (Beine et al., 2001). The Norwegian Polar Institute (NP) is the station owner, and the scientific coordination is done by the Norwegian Institute of Air Research (NILU). The Department of Environmental Sciences and Analytical Chemistry (ACES) at Stockholm University has been measuring the aerosol number size distribution with a closed-loop Differential Mobility Particle Sizer (DMPS) continuously since 2000. The DMPS-

system comprises a custom-built twin DMA setup including one Vienna-type medium DMA coupled to a TSI CPC 3010 covering sizes between 25-800 nm and a Vienna-type short DMA coupled with at TSI CPC 3772 effectively covering sizes between 5 and 60 nm. The number size distributions from the two systems are transferred to a common size grid and then merged. Both systems use a closed-loop setup.

The whole air inlet conforms to WMO/GAW standards (Baltensperger et al., 2003) and EUSAAR
recommendations. The current setup has the inlet drawing a flow of 100 litres per minute (lpm). The inlet is further heated and kept above 0°C to allow gradual evaporation of any droplets or ice crystals as well as to prevent freezing and build-up of ice. Inside the sampling station, the temperature is typically around 20°C.

VRS is located outside 2 km outside Station Nord, a small military airfield on a ~100 km$^2$ fairly flat and ice-free peninsula (Goodsite et al., 2014). The dominating south-westerly winds are caused by the katabatic flow from the
ice cap. In order to minimize the effect of local pollution, the sampling site is placed south-east of the main complex (Heidam et al., 1999, 2004; Nguyen et al., 2013). VRS is located west of the ice stream that floats out from the Arctic Ocean and thus there is both seasonal and multiyear ice right at the doorstep of the station. The aerosol number size distribution in the diameter range of 10 to 900 nm is recorded in 66 bins every 5 minutes with a Scanning Mobility Particle Sizer (SMPS) (Wiedensohler et al., 2012) that is maintained and calibrated by the
Department of Environmental Sciences at University of Aarhus, Denmark. The DMA part of the SMPS is a medium Vienna-type and it is followed by a butanol-based TSI CPC (model 3772).

Alert is the northernmost continuous atmospheric environmental monitoring site in the world. It lies 8 km from the north-eastern shore of Ellesmere Island, which is mostly snow-covered 10 months of the year. The nearby Lincoln Sea typically remains frozen year round. The aerosol number size distribution in the range of 10 to 500 nm
is measured with a TSI 3034 SMPS that is calibrated on site (Leaitch et al., 2013; Steffen et al., 2014).

The measurement site Barrow is located 3 km from the Arctic Ocean, and ~5 km northeast of the centre of the town of Barrow. It is surrounded by rather flat tundra and shallow water bodies. The dominant winds at Barrow are from the eastern sector, which most often bring Arctic marine air mass to the site. The global monitoring division (GMD) at the US National Oceanic & Atmospheric Administration (NOAA) has been measuring aerosols
at Barrow for a few decades, however, only recently a custom-built SMPS, measuring aerosol dry diameters in the range of 10 to 990 nm, has been installed there by the Leibniz Institute for Tropospheric Research (IfT) in Leipzig, Germany.

Tiksi is a town of ~5000 inhabitants in northern Siberia on the shore of Laptev Sea and south of the delta of the Lena river. The aerosol number size distributions are measured at the clean air facility, which is located
approximately 5 km south of town, about 500 m from the shore. There is a 200 m tall hill between the site and town. The dominant local winds are from the western sector, but between April and August light winds from the sea are more common. This was accounted for when choosing the location of the measurement site in order to minimize the contamination of the observations by local pollution. The site, as the other observations sites, is part of the International Arctic Systems for Observing the Atmosphere (IASOA), and it is run by a number of
institutions (for more details see: http://www.iasoa.org). The Finnish Meteorological Institute (FMI) is responsible





for the twin-DMPS system currently covering particle diameters between 3 and 800 nm. The raw data is available for download from the NOAA FTP site (ftp://ftp.etl.noaa.gov/psd3/arctic/tiksi/aerosol). Asmi et al. (2016) provide a complete description of the system setup, inlet and the various routines for assuring the high quality of the data.

## 2.2 Data and quality control

The raw SMPS/DMPS measurements from all sites were corrected for diffusional losses, multiple charging and particle counting efficiency. In order to facilitate direct comparisons between the measurements from the different stations, all integral parameters (e.g. total number, aerosol volume and effective diameter), were calculated over a size range covered by all sites, i.e. 10 to 500 nm. This range includes most of the particles that serve as cloud condensation nuclei (CCN) and, in this environment, most of the particles that scatter and absorb light. Hence,

this size range has strong relevance for climate. All data were recalculated as hourly averages to match the resolution of the back-trajectory analysis that is discussed in Sect. 2.3.

Local pollution may affect the aerosol observations to various degrees at the different sites, although at Zeppelin this is less of an issue (e.g. Hansen et al., 2014). The Alert station is about 7 km NNE of the observatory. Data are filtered for wind directions at the observatory between 0° and 45° (true north) as well as for local events. A

comparison of data at Alert indicates that particles between 20 nm and 100 nm unfiltered by wind direction are 5% higher than the filtered data, and there is no significant difference between filtered and unfiltered data for particles larger than 100 nm (Leaitch: personal communication). At VRS $NO_x$ measurements detect local pollution from the military base or the cars servicing the station, and the corresponding SMPS measurements during local pollution events, which are typically very short, were removed from the dataset (Nguyen et al., 2016). At Barrow

and Tiksi, the risk of contamination of the measurements by pollution from the nearby towns is higher and observations need to be screened carefully. For Barrow, all recorded aerosol data is normally dismissed when wind directions are between 130° and 360° or when the winds are weaker than 0.5 m/s (Polissar et al., 2001). However, we have noticed that despite this, there were still indications of local pollution both in the SMPS and Nephelometer data just after a local wind shift from the potentially "polluted" to the "clean" sector. An extra

prerequisite was thus added, requiring that the winds from the clean sector need to be persistent for at least 24 hours. In addition, data were omitted during the first two hours after the wind shift. The same filters were applied for the Tiksi data, but the polluted sector was confined to the azimuths between 330° and 20°.

While these conditions are useful for limiting the analysis to the background aerosol and facilitating the comparison with the other sites, they inevitably somewhat reduce the temporal and spatial representativeness of

the observations – for some stations more than others. Nevertheless, these datasets provide valuable information about the aerosol characteristics around the Arctic Ocean.

For a more robust statistical analysis, we included all available Arctic DMPS/SMPS observations from recent years. This is because there is no period with overlapping observations from all sites that is long enough. Covering 2.5 to 5 years of observations from each of the sites allows learning about the year-to-year variability without

biasing the results. An example of environmental changes that may correlate with aerosol properties within the observation period, are the minimum and maximum extents of the Arctic Ocean sea-ice (Fig. 1). In September 2012, the Arctic sea-ice coverage reached a record low in modern history (Parkinson and Comiso, 2013). On the other hand , the year of 2013, had the highest mean September ice extent (together with 2009) since 2006 (Serreze





and Stroeve, 2015). Together, these two years capture the last decade's sea-ice extent variability, which is driven by the synoptic conditions and the meteorology (Tilling et al., 2015).

The monthly data availability for all sites is presented in Fig. 2. Each bar shows the fraction of the hourly aerosol measurements that passed the quality control and filtration procedures from the total number of hours in each

month. The observation sites at Zeppelin, Station Nord and Alert have monthly data coverage greater than 50% in two years or more. The site near Tiksi, however, has an annual coverage of ~40%, and the one outside Barrow is left with the poorest data availability (<25% annually) after filtration, but still every month is somewhat represented by measurements from at least one of the years.

### 2.3    Analysis of air-mass back-trajectories

In this study, the Hybrid Single Particle Lagrangian Integrated Trajectory (HYSPLIT_4) model (Draxler and Hess, 1998) was used. The meteorological fields were obtained from the Global Data Assimilation System (GDAS) of NOAA at 1º resolution (http://ready.arl.noaa.gov/archives.php).

A 240-hour 3D back-trajectory was calculated for every hourly aerosol size distribution measured at each of the sites. The receptor altitude was set as 474 m for Zeppelin and 100 m for the other sites. The length of the back-

trajectory calculation was chosen as a balance between the typical lifetime of the aerosols in the arctic troposphere, which is up to two weeks (shorter in summer and longer in winter/spring) for the accumulation-mode particles (Stohl, 2006; Williams et al., 2002), and the increasing uncertainty in the calculation the further back in time it goes. The meteorological parameters along the trajectories were also saved and used for the assessment of their interplay with the aerosol properties. Furthermore, to assess the characteristics of NPF events and to allow a

deeper analysis in a following study, the satellite-derived sea-ice concentrations (http://nsidc.org/data/nsidc-0051) as well as the ocean depth data (http://www.ngdc.noaa.gov/mgg/bathymetry/relief.html) were added as trajectory-related parameters, but only when the trajectories were within the atmospheric mixed layer.

There are various potential sources of error in trajectory calculations. A position error of ~20% of the travelled distance is considered typical (Stohl and Seibert, 1998). One way to estimate the trajectory uncertainty is to use

ensembles of trajectories calculated for the same time and location. Another way is to use multi-particle dispersion models, such as FLEXTRA (Stohl et al., 1998). In this study, however, the large number of single-particle trajectories was sufficient for providing a statistically robust dataset for identifying large aerosol source regions, among other things.

The trajectory analysis used here for each of the sites (and also for all of them at once), was an adaptation to the

one used in Tunved et al. (2013).  This was done in the following way: a concentric coordinate system with the measurement site at its pole was defined around each station. The distance between the centres of neighbouring grid cells was set as 0.5 latitude degrees and 4 longitude degrees. The increased grid cell area further away from the site due to the concentric form, offsets the fact that all trajectories converge to the aerosol site.

The hourly coordinates of the trajectory points were then projected onto the coordinate system, and all the grid

cells, which had one or more trajectory points in them, were considered "hits". This was repeated for all trajectories and the integrated number of hits in each grid cell was divided by the total number of trajectories to provide an estimation of the trajectory probability (i.e. likelihood of the back-trajectories crossing a certain grid cell). On a larger scale, this highlights the main transport pathways of the observed aerosols at a site, for those particles which are not locally originated.



A number of parameters that were derived from the measured aerosol number distribution at each site (e.g. aerosol total volume, number concentration of accumulation-mode particles) or simulated/integrated along the trajectory (e.g. mixing height, distance travelled, integrated precipitation), were compared, and where relevant associated with their corresponding trajectory grid cells.

### 2.4 Clustering the aerosol number size distributions

Aerosol number size distribution exhibits a large degree of spatial and temporal variability, reflecting the variety of processes that has taken place in the air mass before the aerosols were measured. Cluster analysis serves as an excellent method for data-mining. The method relies on the grouping of data to minimize the differences within the data groups, or clusters, while simultaneously maximizing the differences between various clusters. Beddows et al., (2009) demonstrated that the k-means method (Lloyd, 1982) is the most favourable clustering method for aerosol number size distribution data. For this purpose, MATLAB programing tools were used to run the k-means++ algorithm (Arthur and Vassilvitskii, 2007) and to calculate the centroids of the given number of clusters of the aerosol size-segregated number distributions. To allow this, as well as to facilitate the comparison of other aerosol integral properties between the sites, the original observations with the different size ranges and numbers of bins were homogenized and transformed to a common size grid comprising 29 bins equally distributed on a logarithmic scale over the 20 to 502 nm diameter range.

The monthly relative frequency of the different clusters is discussed and linked with the trajectory analysis in order to evaluate the spatial association of the trajectories and the aerosol clusters. However, due to the screening of the Barrow and Tiski dataset based on the local wind directions and their resulting poor data availability, the analysis for these two sites may be biased. The Barrow and Tiksi datasets are therefore excluded from the cluster analysis.

### 3 Results and Discussion

### 3.1 Annual cycle of the aerosol number, surface and volume concentrations

Figure 3 presents the monthly median and 10th to 90th percentile range of the total aerosol (>10 nm) and accumulation-mode (100-500 nm) number concentrations (henceforth $N_{10}$ and $N_{acc}$, respectively), as well as the same percentiles of the aerosol surface area ($S_{10}$) and volume ($V_{10}$) concentrations. All five Arctic sites exhibit an annual cycle with common features, although there are some differences between the sites.

$N_{acc}$ (the solid grey curve in Fig. 3) peaks around April at all sites, with median values between 100 and 200 cm$^{-3}$. The minimum median concentrations, between 20 and 50 cm$^{-3}$, are observed in September or October at all sites except for Tiksi, where the median $N_{acc}$ during these months is around 70 to 80 cm$^{-3}$ and its minimum is below 50 cm$^{-3}$ in July.

The bottom whiskers in Fig. 3 indicate that during the summer and autumn months the air occasionally becomes highly pristine; when $N_{acc}$ drops below 10 cm$^{-3}$ at Zeppelin, Nord and Alert, and below 20 cm$^{-3}$ at Barrow and Tiksi. This coincides with the season of maximum precipitation in the Arctic basin (Serreze and Hurst, 2000), and implies that enhanced wet deposition is a factor in the removal of accumulation-mode particles from the lower Arctic troposphere. Further discussion on the factors that drive the aerosol annual cycle is provided in Sect. 3.5.



The increased precipitation, the destabilization of the lower troposphere as well as the more heterogeneous surface properties (in the melting season) and more variable shortwave radiation in summer can modify the aerosol bulk properties and potentially contribute to an increased spatial and temporal summertime heterogeneity. The large variability in summer as well as the typical particle concentrations at the five sites, are generally in line with the observations made over the Arctic sea further from land, on board the Swedish icebreaker Oden in August and September 1991 (Covert et al., 1996).

The median $S_{10}$ and $V_{10}$, denoted by the orange and blue curves in Fig. 3, respectively, follow the annual cycle of the median $N_{acc}$. They also have the smallest variability and they peak in late spring, as well as increased variability with lower values in summer and autumn. Correlations among these three parameters are driven by the fact that most aerosol number distributions are dominated by accumulation mode particles (see Sect. 3.3.1) and because each of these larger particles contributes more to the aerosol total surface area and volume compared to the Aitken- and nucleation-mode particles. It is important to note though that the $V_{10}$, more than the other bulk parameters, is underestimated due to the exclusion of the particles >500 nm from the integration. At the smaller end of the size distribution, the contribution of particles smaller than 10 nm to the total aerosol surface area and volume, is negligible.

The grey dashed curve of $N_{10}$ in Fig. 3, does not follow the other curves, which follow each other closely. Its most evident feature is the gap that opens between it and the $N_{acc}$ curve in the summer months, and the resulting second peak in $N_{10}$ around July-August. This double-peak in Arctic particle concentration has been observed previously (Croft et al., 2016b; Polissar et al., 2001; Tunved et al., 2013).

The two peaks in $N_{10}$ are of different nature. While the spring maximum is governed by the number of accumulation-mode particles, the summertime peak is due to the increased concentrations of smaller nucleation- and Aitken-mode particles (more in Sect. 3.3.2). It is during the summer months when $N_{10}$ is likely to exceed 500 or 1000 cm$^{-3}$, 10% of the time (depending on the site), as indicated by the top whiskers of the $N_{10}$ curve in Fig. 3.

### 3.2 Annual variations of the aerosol number distributions

The total number concentration, surface area and volume of the particles, do not provide explicit information regarding the shape of the aerosol size distribution. Figure 4 displays the monthly median aerosol number distributions and the interquartile ranges for all five Arctic sites. The general features of the distributions at all locations, month by month, seem quite comparable. However, it could be noticed that in all months except between May and October, the Zeppelin curves (in grey) tend to be below the other curves, i.e. exhibit median lower number concentrations, especially for particles with diameters between 50 and 200 nm.

All monthly Zeppelin distributions exhibit a "Hoppel gap" (Frick and Hoppel, 1993; Hoppel et al., 1986) around 60-80 nm that is more pronounced than in the distributions of the other sites. This implies that the Zeppelin aerosol is more influenced by aerosol-cloud interactions due to its considerably higher elevation than the other sites and proximity to the cloudier north Atlantic air mass. The altitude of Zeppelin may contribute to an increased exposure to free tropospheric air at Zeppelin, with respect to the other low-elevation sites, but the identification of these instances is not trivial and is beyond the scope of this paper.

The distributions from Barrow (in blue) and Tiksi (in pink), on the other hand, have greater particle concentrations than at the other sites in most months, especially in the accumulation-mode range. This is apparently not the result of omitting a subset of the data when the winds arrived from a specific sector, because the filtration of the data





mainly reduced the concentrations of the Aitken-mode particles emitted by local sources. However, the filtration has in fact caused a slight increase in the median concentrations of the accumulation-mode particles at Barrow, but this still not does not change the finding that Barrow and Tiksi have the highest concentration of $N_{acc}$ with or without filtration. It is difficult to separate the contribution of the local pollution from the background aerosol based on the aerosol size distribution alone. The screening, however, seems effective in reducing the measured concentrations of the Aitken-mode particles to their actual background levels.

The monthly medians of Station Nord (orange) and Alert (green) in Fig. 4 show a remarkable resemblance, which makes it hard to separate between the two in some months. These sites are closer to the pole than the other sites, relatively close to each other and are furthest away from the sources of anthropogenic and marine aerosols. The concentrations there are therefore typically lower than at the other coastal and lower latitude sites (Tiksi and Barrow).

Moreover, the shape of the Arctic background aerosol number size distribution at all sites, is dominated by the accumulation mode particles during most of the year, except in summer between June and August. Such distributions normally indicate an aerosol population that is rather "aged" (e.g. Tunved et al., 2013), i.e. measured far from its sources, and possibly, but not necessarily, has been part of one or more cloud cycles before being sampled. It can also mean that there is considerable precursor material associated with their origins.

From Figs. 3 and 4 it is evident that although the general *shape* of the monthly median number size distribution at all sites does not change much during the period the between October and April, the accumulation mode number concentrations have an increasing trend during this period. Figure 4 shows that both inter- and intra-site variability is lowest in April. This suggests that the aerosol properties within the lower Arctic troposphere are rather homogeneous due to the weak aerosol production within the Arctic, inefficient removal processes as well as strong north south transport and vertical stratification in the preceding months. These factors allow the particles to travel further from their sources, mix horizontally and result in increased concentrations and likelihood for the appearance of Arctic Haze around April. In other months, the temporal and spatial variability of the aerosol size distributions is considerably larger.

Later in spring, as conditions become increasingly favourable for NPF events, there is a growing tail of Aitken-mode particles. By July it clearly dominates over the concentrations of the accumulation-mode particles, but that only lasts until September-October, depending on the site. When the Arctic gets dark and the ice sheet is starting to grow again, the NPF signature on the monthly number distributions is mostly lost.

In order to facilitate a quantitative comparison of the monthly aerosol number distributions between the sites, and between other observations, studies, periods, as well as modelling results, the distributions shown in Fig. 4 were described as the sum of three log-normal distributions (Jaenicke and Davies, 1976). Each log-normal distributions is characterized by three parameters; the modal number concentration ($N_i$), the geometrical mean diameter ($D_{g,i}$) and the modal geometrical standard deviation ($\sigma_{g,i}$). In total, there are nine independent fitting parameters that describe each monthly percentile of the aerosol size distribution. Appendix A provides the nine fitting parameters for the median distribution in each of the months and for each site. The fitting parameters for the 10[th], 25[th], 75[th] and 90[th] percentiles can be found in the online supporting material.





### 3.3 Clusters of aerosol number distributions

More than 30,000 hourly aerosol number distributions from Mt. Zeppelin, Station Nord (VRS) and Alert, covering two full years, were used as the input for the k-means analysis, with four output centroids. Each one of the remaining ~30,000 size distributions was ascribed to the cluster whose sum of distances from it was minimal. The

number of clusters was optimized as choosing more than four output clusters resulted in one or more clusters with very few members. In addition, the total sum of the distances from the centroids started levelling out for a greater number of output clusters than the four. The resulting median centroids and the interquartile ranges are shown in Fig. 5 as solid curves and shaded areas, respectively.

Table 1 provides some quantitative information regarding the characteristics of the centroids. The clusters were

numbered by descending aerosol effective diameter, which is useful for comparing between observations and an indication of their age (Croft et al., 2016b).

Three of the four cluster medians in Fig. 5 exhibit an aerosol number distribution that is dominated by accumulation mode particles, with concentrations dropping from 150 to 19 cm$^{-3}$ between clusters 1 and 3, respectively. Thus, the centroid of cluster 1 has the highest total aerosol surface area, volume and mass of

0.21 cm$^2$ m$^{-3}$, $8.9\cdot10^{-7}$ cm$^3$ m$^{-3}$ and 1.34 µg m$^{-3}$, respectively – assuming spherical particles with a density of 1.5 g cm$^{-3}$. Clusters 2 and 3 have decreasing values in all aerosol parameters in Table 1, representing the cleaner conditions, but still with rather aged particles – as indicated by their relatively large effective diameter. The centroid of cluster 4, on the other hand, has a different shape with a mode in the Aitken range and a relatively low effective diameter of 189 nm. The median $N_{10}$ of all number distributions associated with cluster 4 is 273 cm$^{-3}$,

considerably higher with respect to the other clusters, but the median aerosol mass is only 0.31 µg m$^{-3}$ – less than a quarter than the integrated mass of the centroid of cluster 1.

The shaded bands around the centroids in Fig. 5 denotes the inter-quartile range within each cluster for evaluating the spread of the bulk of the distributions within the clusters. It shows little overlapping, which indicates that the clusters are quite distinct from each other. However, the values of the lower and the upper quartiles still lie outside

of the shaded bands. Individual observations can deviate considerably from the centroid of their assigned cluster, but these "outliers" were part of the cluster analysis and affected its output. They are therefore accounted for despite their invisibility in Fig. 5. However, the focus of this work is on the general features of the aerosol number distributions rather than the fine details of individual observations.

#### 3.3.1 Annual variation of the aerosol clusters

The aerosol bulk properties (Fig. 3) and median size distribution (Fig. 4) have a pronounced annual cycle. Although some percentile information is included in those figures, it is not sufficient for understanding the monthly variability of the aerosol size distributions. The clustering of the aerosol size distributions assists in resolving this. Figure 6 shows how the probability of occurrence of an aerosol size distribution assigned to a specific cluster varies between the different months.

There is a common annual pattern in the relative occurrences of the aerosol clusters at the different sites. Most notable is the increasing occurrence of cluster 4 distributions (purple bars) from late spring to late summer – to ~40% at all sites, and the swift drop in September to below ~10%, as the daylight hours rapidly decrease. Alert and Station Nord show more skewed distributions compared to Zeppelin. At Alert and Station Nord, the maximum occurrence of cluster 4 distributions is reached in August rather than in July. This might be due to the





continued retreat of the ice edge in Baffin Bay during the summer (Fig. 1), where most of the cluster 4 trajectories arrive from to Alert and Station Nord (Fig. 7) as well as the stronger exposure to light at more southerly Zeppelin. A closer ice edge allows enhanced concentrations (less diluted) of biogenic aerosol precursor gases such as dimethyl sulphide and an increased probability for NPF events. It should be noted that NPF events were not

confined to only Baffin Bay air masses, as for example (Nguyen et al., 2016) reported a higher chance of observing a NPF event at Nord with southerly air masses arriving from over the Greenland sea. The trajectories that are associated with cluster 4 distributions at Zeppelin according to Fig. 7, mostly arrive from the open North Atlantic. This may be the reason for the closer relation between the irradiance and relative occurrence of cluster 4 distributions (Fig. 6) for Zeppelin, compared with Alert and Station Nord.

Another feature of Fig. 6, common to all sites, is the increase in the monthly occurrence of cluster 1 distributions (indicative of the accumulation mode dominated Arctic Haze) from November to April. The absolute values in April are, however, quite different with ~40%, ~65% and ~90% at Zeppelin, Alert and Station Nord, respectively. A possible explanation for the decreased occurrences of Arctic Haze at Alert compared to Station Nord, which does not seem to be due to year-to-year variability, are the frequent katabatic winds from the high mountains that

pull mid tropospheric and less polluted air to the site at Alert (Morin, 2005). This is clearly seen in day to day fluctuations of ozone depleted boundary layer air and un-depleted ozone containing free tropospheric air at Alert after polar sunrise (Barrie et al., 1988, 1994).

For Zeppelin, on the other hand, there is a fair number of trajectories arriving from the North Atlantic in April that are not associated with cluster 1 distributions. In addition, according to the trajectory analysis, the air masses

arrive, on average, from higher elevations compared to Station Nord, as Zeppelin is sometimes above the regional mixed layer. This results in a relative low frequency of Arctic Haze at Zeppelin in April, compared to Station Nord and Alert.

The mean annual relative occurrence of the clusters, indicated at the top of panels in Fig. 6, provide a general view on the year-to-year variability. Only the years with an annual coverage >55% are included to minimize

biases. However, the Zeppelin dataset is missing April 2015 and July 2012 (Fig. 2), which results in an underestimation of ~3-4% in the occurrence of clusters 1 and 4 in the respective years (~40% average monthly occurrence divided by 12 months). There are systematic differences between the sites that are not a result of the year-to-year variability. These include the greater frequency of cluster 3 and 4 distributions at Zeppelin, as well as cluster 1 at Station Nord.

Barrow and Tiksi are excluded from this specific analysis and discussion due to the wind-direction-based data filtration required due to the local particle sources (see Sect. 2). This preferentially dismisses parts of the dataset associated with transport form certain sectors with varying representation through the year. For example, the North Pacific flow to Barrow, which is more frequent in the spring and summer (see Fig. 12). This filtration increases the uncertainty in the monthly relative occurrence and annual frequency and does not allow an unbiased

comparison with the other sites.

### 3.3.2    The relationship between the atmospheric flow and the aerosol clusters

The trajectory analysis – as described in Sect. 2.3, was done to identify any relationship between the aerosol properties recorded at the various sites and the geographical positions of the sampled air masses in the preceding days. Each subplot in Fig. 7 shows a trajectory probability map for a given site and aerosol cluster. It indicates





that the trajectory-occurrence density field is typically far from isotropic, i.e. there are preferred pathways to each site. Trajectories associated with cluster 1 distributions (highest $N_{acc}$) at Zeppelin, for instance, are far more likely to arrive from the eastern sector than from the western sector. The trajectory frequency of cluster 2 (middle $N_{acc}$) distributions at Station Nord is another example; two main branches are separated by Ellesmere Island. The

southern branch mostly follows the western coast of Greenland (an ice sheet that is 2 to 3.2 km in altitude), and this demonstrates the channelling effect of the topography on the flow in that region.

A closer examination of the green and yellow shades in Fig. 7, as well as the mean trajectory positions (red curves) for each site separately, reveals a counter-clockwise rotation of the air mass origin when moving from the cluster 1 aerosol distributions with their high aerosol effective diameters, to the relatively "fresh" cluster 4 members.

This applies for all three sites, indicating that an aerosol size distribution of cluster 1 is more likely to arrive from the Russian side of the Arctic Ocean, while trajectories from North of the Canadian arctic are more likely to be associated with cluster 3 (lowest $N_{acc}$) distributions.

The trajectory densities that are associated with cluster 4 (recent NPF event) aerosols at the three main aerosol sites have major southerly branches (Fig. 7) that come either from the North Atlantic or Baffin Bay – although

there is still some contribution from intra-Arctic flow, mostly over shallow-waters close to the shoreline. Elevated DMS fluxes from the open water compared to the frozen ocean (Lana et al., 2011; Leck et al., 2002; Mungall et al., 2016) and ammonia fluxes from coastal bird colonies (Croft et al., 2016b; Wentworth et al., 2016) may be an important factor (Croft et al., 2016a). It is also evident that the trajectories of cluster 4 distributions have an extremely low probability of passing over the more polluted Asian or the European mainland during the preceding

10 days.

The areas where the NPF events take place are much closer to the sites with respect to the extent of the shaded areas for the cluster 4 distributions, at the bottom of Fig. 7. This is because the coloured shades cover the area of the full 10-day trajectories, while the newly formed aerosols are likely to grow into cloud condensation nuclei (CCN)-sized particles (>~70 nm) in a day or two (Dal Maso et al., 2005; Kulmala et al., 2001). At high-latitude

sites including the Arctic it may take up to three days, due to a mean growth rate of newly formed particles of as low as 1 nm hr$^{-1}$ (Asmi et al., 2016; Kulmala et al., 2004; Ruuskanen et al., 2007; Ström et al., 2009; Tunved et al., 2003), unless close to the surface of the open sea in the Arctic where such growth may occur over a few hours (Willis et al., 2016). A large part of the area covered by cluster 4 trajectories is where the air mass may have been exposed to marine and coastal precursor gases and where the condensation sinks are small due to low pre-existing

particle concentrations, both of which enhance the probability of NPF.

### 3.4 Source regions of accumulation-mode particles

Arctic Haze is characterized by elevated concentrations of light scattering and absorbing accumulation-mode particles. The residence time of a particle in the accumulation-mode diameter range, is rather long with respect to the smaller or larger size ranges – especially when there is low precipitation such as in the winter/spring Arctic

air mass (Barrie, 1986). This means that accumulation-mode particles can travel great distances in the Arctic and be traced back to their source regions.

Figure 8 shows by site, the spatial distribution of median concentrations of accumulation mode particles in the trajectory grid cells. It was derived from the $N_{acc}$ concentrations observed at the sites at the time of the air mass arrival combined with the origin of air mass based on back trajectory analysis. Each site's entire dataset is included





(Fig. 2). To reduce uncertainties, the shaded areas in each panel include only grid cells that were crossed by at least five trajectories. Displaying the median $N_{acc}$ is useful for observing the common features of its spatial distributions, because it is not affected by extreme cases (like the mean value).

The median $N_{acc}$ values at Tiksi and Barrow are higher with respect to the other sites, as indicated by the greater

extent of green and yellow shades. Zeppelin is most affected by continental Europe, although these trajectories are not associated with the highest median $N_{acc}$. The regionally elevated Zeppelin $N_{acc}$ values, associated with North Atlantic trajectories, are possibly due to the contribution of the sea salt particles (Glantz et al., 2014).

The similarities between the highlighted regions in all Fig. 8 panels indicate that the Arctic sites share the Asian side of the Arctic as the main large-scale source region of accumulation mode aerosols. This is consistent with

modelling studies of Arctic haze transport (e.g. Barrie et al., 1989; Christensen, 1997; and many others). Analyses involving observational data, modelling and or emissions (Hirdman et al., 2010b; Sharma et al., 2006) highlight the same regions as potential sources of black carbon for the Arctic sites.

According to Fig. 8, Zeppelin and Tiski have an additional source region for accumulation-mode particles from western Russia and western Kazakhstan below 50° N – the area to the north of the Black, Caspian and Aral Seas.

The eastern part of this area is one of the global hot-spots for desert dust (Engelstaedter and Washington, 2007), which smaller particles are in the accumulation mode size range (e.g. Mahowald et al., 2014). Trajectories from this region apparently barely reach the other sites within the ten days' frame of the trajectory analysis.

The median accumulation-mode particle concentration over the Arctic Ocean is lower than that over land, because it is frequented more by trajectories associated with lower concentrations (higher cluster numbers) that also pass

over the ocean (Fig. 7).

It is also important to note that when high $N_{acc}$ occur near the edge of the analysed domain in Fig. 8, it is possible that the source regions are farther away. It may also be possible that some of these regions are "in the shade" or "behind" the actual source regions, so the trajectories first pass over those regions before reaching the source regions. However, without trajectories that cover the "shaded" regions and avoid the actual source regions, it is

not possible to separate those regions using this methodology. As the durations of high quality observations increase, these analyses will reveal even more clear results with greater accuracy.

To avoid further filtering and to improve the statistics, the results of the trajectory analysis shown in Fig. 8 do not account for the altitude of the trajectory above the surface. This means that the median $N_{acc}$ was derived from all trajectory heights for each of the grid cells. To ensure that this does not affect the results of the analysis, similar

maps were derived for two subsets of trajectory points: (i) those that were within twice mixing level height according to the meteorological dataset; and (ii) those above that – representing the planetary boundary layer (PBL) and free troposphere (FT), respectively. The distributions did not show any considerable difference (not shown here). This may be due to the increasing uncertainties in the trajectory height with time, which makes in many cases the determination of whether a single-particle trajectory, which is a few days old, is within the PBL

or in the FT very unreliable. For that, running trajectory ensembles with different methods for treating vertical motions may be beneficial, but it is not within the scope of this analysis.

Instead, panel (a) in Fig. 9 shows the map of the mean trajectory altitude for the combined dataset – containing all trajectories from all five sites. Panel (b) displays the median $N_{acc}$ concentrations derived from this dataset. They show that over central Siberia and western Russia, where the median $N_{acc}$ is rather high, the mean trajectory

altitude is around 1000 m above the surface. This reinforces the claim that these areas are source regions for





accumulation-mode particles in the Arctic. The area to the south and east of lake Baikal is also highlighted with high $N_{acc}$ concentrations, but the mean trajectory altitude there is more than 2000 m, which may be an indication that the aerosols are originated from sources further away. For example, Huang et al. (2015b) showed that Asian dust occasionally reaches the Arctic region, and (Liu et al., 2015) identified the main source of their observed

black carbon in the lower troposphere of the European Arctic as the region between 50 and 60 degrees north in Asia.

Additional information about the variability can be obtained by plotting and comparing higher and lower percentile values, rather than only the median or mean. However, sometimes the extreme cases are those of interest and they show up in the map displaying the maximum values (not shown here). One such case occurred around

July 11[th], 2015, which brought heavy pollution from Central Alaska to Zeppelin, after the occurrence of an exceptionally high number of forest fires in the preceding weeks[1]. The associated extreme $N_{acc}$ values considerably affect the grid cell mean value, and thus demonstrate why keeping the full information is useful for avoiding misinterpretation of the results.

### 3.5   Main drivers of the annual cycle of $N_{acc}$

During the course of a year, the Arctic environment undergoes various changes with the potential to affect local and regional aerosol properties. In a modelling study, (Croft et al., 2016b), identified wet removal by snow or rain as the main sink for accumulation-mode particles. Condensation (including cloud processing) and transport were found to be as the main sources of these particles. As modelled, all these processes were most active in the summer months. Their results are consistent with observed Arctic precipitation distributions although there is considerable

variability and uncertainty in the precipitation amounts and distributions in the Arctic (Serreze and Hurst, 2000). There is general agreement that most of the Arctic receives more precipitation in summer than in winter, except for the North Atlantic sector (including the Spitzbergen region). This is due to increased moisture and heat fluxes from the sea when sea ice retreats, which favours cloud formation and precipitation and the lack of moisture in very cold Arctic air masses.

For wet removal of aerosols, precipitation along the trajectory ($P_{traj}$) is more relevant than the mean monthly precipitation around the measurement site itself. This information can be derived from the meteorological fields used for the trajectory analysis. Although there is a large uncertainty in $P_{traj}$, for individual days, combining many days allows for a more accurate estimate of the monthly median. Panel (a) in Fig. 10 shows that the peak median accumulated precipitation along the 240-hr back-trajectories is around August for all sites except for Zeppelin –

where this peak is in September (the dark season values at Zeppelin are considerably higher too). This is a couple of months later than the maximum monthly local precipitation (Serreze and Hurst, 2000) – possibly and partly due to the change in the transport patterns (panel (c) in Fig. 10) and the continued melting of the sea ice as well as changes in available moisture as air temperatures and absolute humidity rise. On average, the air masses spend more time in August than in June at lower/wetter latitudes over open waters, potentially increasing $P_{traj}$.

[1]The URLs for the fire map of the period 30 June to 10 July, 2015, and a post by the Alaska Division of Forestry from 1 July 2015:

https://lance.modaps.eosdis.nasa.gov/imagery/firemaps/firemap.2015181-2015190.2048x1024.jpg
https://akfireinfo.com/2015/07/01/could-the-2015-alaska-fire-season-top-the-record-year-of-2004/





The trajectory-precipitation maximum is nearly coincident with the lowest monthly $N_{acc}$ concentrations (Fig. 3), and they are anti-correlated. This suggests that wet removal is the main driver of the $N_{acc}$ annual cycle, or at least an important contributor to atmospheric lifetime of accumulation-mode particles in the autumn. However, a more complex picture emerges when analysing the fitting parameters, $a$ (slope) and $b$ (intercept), in the following

regression - Eq. (1):

$$\log(N_{acc}) = a \cdot X_{traj} + b, \tag{1}$$

where $X_{traj}$ denotes the median value of a trajectory-derived parameter, such as $P_{traj}$, or any of the others that are

shown in Fig. 10.

The slope $a$ in Eq. (1) is the *relative* change in $N_{acc}$ for every additional unit of $X_{traj}$, i.e. $dlog(N_{acc})/dX_{traj}$. Panels (a) through (d) in Fig. 11 displays the values of $a$ (adjusted to make the units more intuitive) for the corresponding parameters shown in Fig. 10. Only statistically significant values ($P<0.05$) are shown.

Negative values in Fig. 11a indicate that $N_{acc}$ is reduced with increasing precipitation – as expected if wet removal

is the main process controlling the concentration of the accumulation-mode particles. While this claim appears to be valid for Zeppelin for most of the year (8 out of 12 months), it seems to be the opposite for Tiksi, i.e. $N_{acc}$ rises with increased trajectory precipitation. It is certainly not a causal relation – it just means that samples with greater $N_{acc}$ are associated with higher $P_{traj}$. In a storm, for instance, transport and precipitation may be coincidental. Even if the scavenging is reasonably efficient, the residual transported aerosol may be sufficient to give a positive

association of $N_{acc}$ and $P_{traj}$. Alternatively, the addition of the particles to the sampled air mass may sometimes take place *after* most precipitation fell.

The other sites, however, show a weak negative tendency, but the relative effect of wet deposition is lowest between June and October – when it precipitates the most. This may be surprising, but it could be partly due to the high absolute uncertainty in the individual $P_{traj}$ values– especially in the summer months when there are more

convective clouds, which are not resolved by the coarse meteorological fields. However, higher median summertime $P_{traj}$ suggests a greater potential for aerosol wet scavenging on a regional scale.

Panels (b), (c) and (d) in Figs. 10 and 11 explore other trajectory-derived potential candidates for playing a key role in the $N_{acc}$ annual cycle and or its monthly median concentrations. $X_{traj}$ in Eq. 1 is replaced by the time the trajectory spent in cloud ($T_{cloud}$), time in the planetary boundary layer ($T_{PBL}$) and the trajectory length ($Dist$) in

panels (b), (c) and (d), respectively.

Figure 10b indicates that $T_{cloud}$ is minimal for all sites around June. This may not seem in line with the observations of increased summertime cloudiness in the Arctic. But the vertical dimension should be accounted for in such a comparison as well as the fact that parts of the trajectories may be outside the Arctic region. There are additional causes for the apparent discrepancy between satellite- and reanalysis-derived annual cycle of Arctic cloudiness

which are explained in Chernokulsky et al. (2012). Nevertheless, the results of the trajectory analysis shown in Fig. 10b, suggest that in the dark season, there is more available time for cloud processing, which acts to increase the aerosol sizes while not considerably affecting their number concentrations in non-precipitating clouds.

The opposite phases of $P_{traj}$ and $T_{cloud}$ (cf. panels (a) and (b) in Fig. 10) indicate that in the summer and spring months, more precipitation falls for every hour spent in a cloud, compared to the dark season. According to Fig.

11b, the little time spent in precipitating clouds between June and October, still tends to lower $N_{acc}$, but in the dark





season the extended cloud processing and low precipitation rates are associated with slightly increased $N_{acc}$ concentrations.

Figure 10c shows the annual cycle of the median $T_{PBL}$. The rationale is that most of the sources of the aerosols and their precursors are within the PBL, but higher $T_{PBL}$ are not more common in the polluted season hence cannot

explain the annual cycle of $N_{acc}$. Actually in Tiksi, and to a lesser extent Alert, there is a greater PBL influence in the summer/spring months – rather than the opposite. However, the relative change in $N_{acc}$ for every additional hour spent in the PBL (Fig. 11c) is negative for most of the sites and especially between May and October. This is an indication that the PBL is a net sink for $N_{acc}$ during these months – probably mainly due to wet removal.

Another trajectory-derived parameter is *Dist*, which is also a measure of mean wind speed along the 10-day

trajectory. Figure 10d indicates that this distance is minimal in late spring/early summer between May and July for all sites and maximal in early winter December-January (Tiksi's cycle is less pronounced). This means that in early winter there is potential for transporting aerosols from greater distances compared to late spring/early summer. Figure 11d does not provide a consistent picture of whether greater distances are linked with lower or higher concentrations of accumulation-mode particles on a monthly scale. This is because it does not contain

information from where exactly the air is arriving during the different seasons and whether it travels over the source regions of the accumulation-mode particles. This information is shown in Fig. 12.

The top maps in Fig. 12 focus on the differences between the main summer and winter months. It shows that in June and July (the pink shade) the sampled air at all sites is rather restricted to the Arctic Ocean, with flow over the Canadian Arctic and Greenland that reaches only Alert, Station Nord and Zeppelin. There is an occasional

intrusion of North Pacific maritime air through the Bering Strait in the summer months. In December and January, however, the trajectory areal coverage is greater for Alert, Station Nord and Zeppelin, and all sites indicate an expansion into the Asian side of the Arctic, with Zeppelin having some north European influence. This means that in wintertime, as the polar vortex expands southward, the 10-day trajectories are much more likely to travel over regions with anthropogenic influence compared to the summer months, when the Arctic is more isolated.

The bottom maps in Fig. 12 display the trajectory coverage in spring versus autumn, but the differences are less pronounced with respect to the top maps. This is because the annual cycle of the median trajectory length has its extremes in winter and summer (Fig. 10d). However, with respect to the geographical coverage, the spring trajectories are more likely to arrive from lower latitudes than in autumn, and hence still have the potential to carry more accumulation mode particles. In effect, the north does not experience a spring but rather winter, short

summer and a short autumn.

The median $N_{acc}$ continues to rise from winter to spring at all sites (Fig. 3) despite the slight reduction in the Asian influence, causing a "phase shift" between "maximum" transport (December-January) and highest annual median $N_{acc}$ (April). This suggests that transport alone cannot explain the $N_{acc}$ cycle in the Arctic. However, a simple conceptual "box" model that considers the Arctic dome, bounded by the Polar vortex, as an aerosol reservoir (or

a mixing chamber). The same mass of air comes in and out of the dome due to continuity, but if more particles enters the dome (both from the outside as well as those produced/emitted inside) compared with the number of particles that exit the dome and removed within it, the particle concentration inside the dome would increase. This seems to be the case between January and April; although the Arctic dome is shrinking and thus excluding some of the potential anthropogenic sources from the dome, the *net $N_{acc}$* flux into the Arctic is still positive due to the

low removal rate. Around May, when most of the winter and early spring $N_{acc}$ sources are outside of the shrinking



Arctic dome, when at the same time wet scavenging increases (Fig. 10), the regional aerosol concentrations within the dome start to decrease.

Another point to consider is the mixing between the transported air mass and its surroundings during the transport. This is required when running single-particle trajectories for obtaining a better representation of reality. In that case, the mixing is the reason why two identical trajectories that do not pass near any $N_{acc}$ sources, and even without any clouds or precipitation along them, are expected to have higher $N_{acc}$ in the Arctic in April versus October or January. Without profound aerosol dynamics, as the trajectories extend in time, the initial particle concentration is less important, and the mixing with the surrounding air plays an increasingly important role. This is most strongly indicated at the high Arctic sites of Alert and Nord, by the relatively small inter-quartile variability and high 25$^{th}$ percentile of $N_{acc}$ in Fig. 3.

The analysis discussed in this section explores the effects of various trajectory-derived global parameters on the observed $N_{acc}$ at the end-point of the trajectory one by one. It may be possible to perform a more detailed study that accounts for the various processes simultaneously and iteratively with the same dataset, and it would be beneficial to use one or more observed aerosol distributions along the trajectory. For instance, an analysis of trajectories passing Alert and then Nord and or Zeppelin would potentially reveal or support some of the discussed results. This might be done in a follow-up study.

The results reported above suggest that on a larger scale the patterns of the airflow (transport and mixing), seasonal precipitation and their link to aerosol processes and source regions are the main drivers of the pronounced annual cycle of the accumulation-mode aerosols observed in the Arctic. Wet removal plays an increasingly important role in reducing $N_{acc}$, regionally and locally, from spring to autumn. However, the small amount of precipitation in winter and early spring, the long time that the air spends in clouds and the extension of the polar vortex, allow average regional concentrations of the accumulation-mode particles to increase during this period.

It is also worth noting that the typical lifetime of the accumulation-mode aerosols is on average shorter in summer and autumn with respect to winter and spring due to the increased accumulated precipitation along a ten-day back-trajectory (Fig. 10a). This means that in summer/autumn a shorter period than 240-hr could be used for the back-trajectory calculation to determine the areal extent with a potential to considerably affect the aerosol size distribution. In winter/spring more than 240 hours may be needed. This would make the differences between the shaded areas in Fig. 12 even more pronounced. This further emphasises the important role of the large-scale flow in determining the properties of the aerosols throughout the year.

## 4 Summary and Conclusions

This paper discusses the annual cycle of several physical aerosol properties at five sites around the Arctic Ocean, with the focus on the accumulation-mode particles. In order to perform such task and to allow comparison among the sites, multi-year observations of the aerosol number size distributions were collected and compiled and the dataset was quality-controlled and homogenized with hourly aerosol concentrations that covers the diameter range of 20 to 500 nm in 29 bins.

The cluster analysis revealed four distinct aerosol number size distributions that were reconcilable with a mixture of aerosol dynamical processes and long-range transport and removal processes. One of these was a number size distribution characteristic of a recent new particle formation event in relatively clean air, which prevails mostly





in June to August. These four modes of aerosol constitute an important diagnostic that climate models need to reproduce if they are accurately representing climate active aerosols.

On the large scale, all sites showed pronounced annual cycles with common features. The total aerosol surface area, volume and accumulation-mode concentration increased through the winter, peaked in April and decreased
to a minimum around autumn. This is due to the interplay of a number of factors, with the large-scale flow, wet removal by rain and snow and cloud processing being the most important. The total number of aerosols experienced a second peak in late summer in addition to the spring peak. This was due to rather common regional events of new particle formation, which produced small particles in relatively high numbers. These events are commonly related to incoming shortwave radiation, low condensation sink and the accumulation of precursor
gases prior to the nucleation event. This mostly took place over the Arctic Ocean and North Atlantic Ocean with some indication of association with coastlines and the ice edge.

The prevailing weather and the associated air-mass trajectories on a temporal scale of about a week determine the aerosol monthly characteristics in a specific year. This is because it affects the environmental conditions such as the ice extent, the regional heat and moisture fluxes and the biological activity, and hence the aerosols and the
clouds, but there are consistent differences among the sites that are beyond the year-to-year variability. They are related to the location of the measurement site. For example, Alert is affected by frequent katabatic winds from nearby hills, which during winter/spring frequently draw free tropospheric air to the site. In a stable and stratified Arctic troposphere, this could mean the observation of very different aerosols compared to the regional surface based-inversion layer of the Arctic air mass.

Another example is the fact that the Zeppelin site is more than 400 m high above the nearby sea, which is open year round – unlike the sea near the other sites (Fig. 1). This results in a difference in the cloud and precipitation patterns and hence the lowest background concentrations of accumulation-mode particles in April, compared to the other sites included in this study. On the other hand, its proximity to Europe (and rather frequent ship traffic) makes it more likely for some pollution to make it to the site during the cleaner months of summer and autumn.

There is no single site that can be considered as fully representative for the entire Arctic region with respect to aerosol number concentrations and distributions. It is therefore important to understand which processes cause the differences between the sites and to test how well state-of-the-art aerosol models can capture these effects. The multi-site aerosol dataset could also be used for looking into how the aerosol number size distribution is altered when the air flows from one site to another.

It is expected that with a continued trend of reduction in Arctic sea ice, the emissions of biogenic sulphur gases that are aerosol precursors and hence affect aerosol growth and formation would increase in summer. This would alter the CCN properties and thus the clouds in the region. It is not clear whether this would result in a positive (Levasseur, 2013) or a negative (Gabric et al., 2005) feedback to the ongoing Arctic warming. This is because of the complex interactions and feedbacks between the aerosols, the clouds, the longwave and shortwave radiation,
the ocean dynamics, the biota and the environment (Browse et al., 2014). Also, the potential for increased shipping emissions and other Arctic industrialization will make this highly uncertain. Eventually, an improved understanding of these interactions would reduce the uncertainties in future projections of Arctic climate changes and its implications for the rest of the world.





**Appendix A**

An observed aerosol number size distribution can often be well-approximated as the sum of three lognormal distributions. Each lognormal distribution is described with three parameters: $N$, $\sigma$ and $\overline{D_p}$ [nm], which denote the particle concentration [cm$^{-3}$], the geometric standard deviation [dimensionless] and the mean geometric diameter

5  [nm], respectively.

The sum of the three log-normal distributions is calculated in the following way :

$$n_N^\circ(\log D) = \sum_{i=1}^3 \frac{N_i}{(2\pi)^{1/2} \log \sigma_i} exp\left(-\frac{(\log D_p - \log \overline{D}_{pi})^2}{2 \log^2 \sigma_i}\right) \tag{A1}$$

10  The tables below provide the fitting parameters of the three log-normal representations of the monthly median aerosol number distribution for each of the Arctic sites and each of the months, shown in Fig. 4. The fitting parameters for the 10[th], 25[th], 75[th] and 90[th] percentiles are given in the online supporting material.

| ALERT | Mode 1 | | | Mode 2 | | | Mode 3 | | |
|---|---|---|---|---|---|---|---|---|---|
| | $N$ | $\log \sigma$ | $\overline{D_p}$ | $N$ | $\log \sigma$ | $\overline{D_p}$ | $N$ | $\log \sigma$ | $\overline{D_p}$ |
| Jan | 27.47 | 0.269 | 62 | 57.94 | 0.183 | 176 | 3.63E+00 | 0.093 | 200 |
| Feb | 32.37 | 0.275 | 70 | 61.71 | 0.172 | 178 | 6.85E-01 | 0.066 | 385 |
| Mar | 53.51 | 0.288 | 70 | 77.19 | 0.174 | 178 | 1.02E-05 | 0.266 | 265 |
| Apr | 58.58 | 0.274 | 68 | 109.65 | 0.172 | 174 | 2.01E-06 | 0.272 | 279 |
| May | 28.45 | 0.210 | 44 | 69.97 | 0.189 | 132 | 1.68E+01 | 0.139 | 214 |
| Jun | 51.92 | 0.205 | 36 | 51.15 | 0.188 | 108 | 6.25E-01 | 0.118 | 277 |
| Jul | 96.33 | 0.234 | 52 | 7.08 | 0.093 | 138 | 4.48E+00 | 0.125 | 200 |
| Aug | 62.80 | 0.168 | 29 | 56.03 | 0.227 | 86 | 2.16E-06 | 0.281 | 336 |
| Sep | 31.11 | 0.230 | 30 | 22.07 | 0.170 | 116 | 2.81E-01 | 0.080 | 336 |
| Oct | 13.30 | 0.260 | 49 | 20.85 | 0.165 | 141 | 6.41E-01 | 0.100 | 312 |
| Nov | 30.22 | 0.263 | 63 | 51.52 | 0.175 | 172 | 8.82E-02 | 0.041 | 443 |
| Dec | 34.64 | 0.288 | 70 | 49.54 | 0.178 | 177 | 4.78E-09 | 0.250 | 313 |

| BARROW | Mode 1 | | | Mode 2 | | | Mode 3 | | |
|---|---|---|---|---|---|---|---|---|---|
| | $N$ | $\log \sigma$ | $\overline{D_p}$ | $N$ | $\log \sigma$ | $\overline{D_p}$ | $N$ | $\log \sigma$ | $\overline{D_p}$ |
| Jan | 44.29 | 0.246 | 63 | 56.71 | 0.162 | 132 | 8.59E+01 | 0.199 | 200 |
| Feb | 62.24 | 0.324 | 70 | 122.54 | 0.212 | 143 | 5.19E+01 | 0.254 | 200 |
| Mar | 78.57 | 0.362 | 70 | 73.87 | 0.165 | 126 | 9.11E+01 | 0.189 | 200 |
| Apr | 34.76 | 0.260 | 59 | 95.75 | 0.159 | 128 | 1.05E+02 | 0.181 | 200 |
| May | 18.40 | 0.156 | 46 | 128.48 | 0.147 | 121 | 1.29E+02 | 0.168 | 200 |
| Jun | 47.27 | 0.207 | 40 | 61.41 | 0.155 | 114 | 1.65E+01 | 0.178 | 200 |



| | N | log σ | $\overline{D_p}$ | N | log σ | $\overline{D_p}$ | N | log σ | $\overline{D_p}$ |
|---|---|---|---|---|---|---|---|---|---|
| Jul | 199.44 | 0.296 | 58 | 9.89 | 0.118 | 147 | 4.73E+00 | 0.187 | 246 |
| Aug | 102.72 | 0.160 | 36 | 45.50 | 0.137 | 102 | 2.17E+01 | 0.185 | 200 |
| Sep | 13.14 | 0.163 | 30 | 60.44 | 0.180 | 105 | 1.13E+01 | 0.290 | 201 |
| Oct | 8.42 | 0.168 | 29 | 45.17 | 0.203 | 116 | 1.79E+01 | 0.264 | 200 |
| Nov | 30.47 | 0.362 | 42 | 88.09 | 0.174 | 141 | 2.52E+01 | 0.140 | 265 |
| Dec | 20.16 | 0.215 | 42 | 89.12 | 0.200 | 129 | 1.02E+02 | 0.223 | 200 |

| STATION NORD | Mode 1 | | | Mode 2 | | | Mode 3 | | |
|---|---|---|---|---|---|---|---|---|---|
| | N | log σ | $\overline{D_p}$ | N | log σ | $\overline{D_p}$ | N | log σ | $\overline{D_p}$ |
| Jan | 48.23 | 0.362 | 64 | 39.40 | 0.168 | 175 | 1.56E+01 | 0.362 | 200 |
| Feb | 42.20 | 0.311 | 49 | 49.45 | 0.176 | 161 | 1.78E+01 | 0.247 | 200 |
| Mar | 46.09 | 0.281 | 45 | 103.95 | 0.206 | 161 | 2.78E+00 | 0.078 | 204 |
| Apr | 50.76 | 0.230 | 43 | 175.74 | 0.203 | 160 | 1.62E+00 | 0.065 | 237 |
| May | 38.21 | 0.192 | 37 | 97.90 | 0.220 | 138 | 3.76E+00 | 0.090 | 233 |
| Jun | 78.89 | 0.341 | 39 | 17.79 | 0.146 | 121 | 4.13E+00 | 0.142 | 239 |
| Jul | 105.81 | 0.263 | 41 | 14.34 | 0.136 | 127 | 4.23E+00 | 0.144 | 229 |
| Aug | 116.58 | 0.262 | 31 | 11.29 | 0.112 | 117 | 4.91E+00 | 0.138 | 200 |
| Sep | 60.44 | 0.334 | 26 | 23.13 | 0.165 | 122 | 1.49E+00 | 0.152 | 286 |
| Oct | 29.10 | 0.306 | 39 | 26.57 | 0.149 | 128 | 3.95E+00 | 0.140 | 252 |
| Nov | 39.97 | 0.315 | 51 | 62.00 | 0.178 | 172 | 1.78E+00 | 0.362 | 450 |
| Dec | 35.49 | 0.362 | 50 | 51.06 | 0.187 | 171 | 5.65E-01 | 0.362 | 450 |

| TIKSI | Mode 1 | | | Mode 2 | | | Mode 3 | | |
|---|---|---|---|---|---|---|---|---|---|
| | N | log σ | $\overline{D_p}$ | N | log σ | $\overline{D_p}$ | N | log σ | $\overline{D_p}$ |
| Jan | 86.53 | 0.277 | 70 | 74.43 | 0.192 | 202 | 1.91E-08 | 0.201 | 296 |
| Feb | 116.54 | 0.304 | 70 | 90.36 | 0.202 | 179 | 2.48E-08 | 0.073 | 359 |
| Mar | 148.31 | 0.362 | 70 | 101.00 | 0.184 | 169 | 5.29E+00 | 0.072 | 244 |
| Apr | 104.63 | 0.287 | 53 | 153.52 | 0.182 | 168 | 1.15E-06 | 0.254 | 287 |
| May | 45.09 | 0.319 | 50 | 64.77 | 0.194 | 168 | 2.24E+00 | 0.081 | 258 |
| Jun | 103.25 | 0.187 | 37 | 91.32 | 0.171 | 104 | 1.38E+01 | 0.144 | 200 |
| Jul | 110.63 | 0.207 | 37 | 32.89 | 0.153 | 122 | 3.30E+00 | 0.185 | 200 |
| Aug | 110.68 | 0.227 | 43 | 60.88 | 0.175 | 150 | 9.28E-01 | 0.090 | 436 |
| Sep | 94.34 | 0.200 | 45 | 21.48 | 0.119 | 131 | 3.17E+01 | 0.149 | 200 |
| Oct | 78.11 | 0.235 | 47 | 60.41 | 0.183 | 173 | 1.03E+01 | 0.090 | 214 |
| Nov | 57.27 | 0.291 | 46 | 63.36 | 0.224 | 167 | 1.07E+01 | 0.118 | 216 |
| Dec | 76.42 | 0.303 | 68 | 7.82 | 0.098 | 165 | 4.96E+01 | 0.177 | 220 |





| ZEPPELIN | Mode 1 | | | Mode 2 | | | Mode 3 | | |
|---|---|---|---|---|---|---|---|---|---|
| | $N$ | $\log \sigma$ | $\overline{D_p}$ | $N$ | $\log \sigma$ | $\overline{D_p}$ | $N$ | $\log \sigma$ | $\overline{D_p}$ |
| Jan | 19.61 | 0.362 | 41 | 4.47 | 0.069 | 238 | 3.51E+01 | 0.176 | 200 |
| Feb | 30.14 | 0.362 | 46 | 38.53 | 0.132 | 194 | 1.59E+01 | 0.215 | 200 |
| Mar | 38.39 | 0.362 | 59 | 54.30 | 0.138 | 193 | 1.89E+01 | 0.234 | 200 |
| Apr | 40.15 | 0.233 | 39 | 68.99 | 0.213 | 151 | 3.26E+01 | 0.141 | 202 |
| May | 105.53 | 0.263 | 33 | 37.79 | 0.139 | 120 | 5.76E+01 | 0.156 | 200 |
| Jun | 102.24 | 0.222 | 35 | 47.36 | 0.145 | 115 | 2.27E+01 | 0.142 | 200 |
| Jul | 147.84 | 0.266 | 41 | 25.75 | 0.119 | 123 | 1.75E+01 | 0.131 | 200 |
| Aug | 102.39 | 0.220 | 35 | 32.98 | 0.164 | 121 | 4.67E+00 | 0.196 | 200 |
| Sep | 43.14 | 0.255 | 34 | 22.98 | 0.186 | 120 | 1.74E+00 | 0.243 | 204 |
| Oct | 16.07 | 0.266 | 39 | 19.24 | 0.174 | 150 | 4.24E+00 | 0.301 | 200 |
| Nov | 14.25 | 0.271 | 33 | 31.80 | 0.218 | 165 | 2.34E+00 | 0.082 | 250 |
| Dec | 14.91 | 0.276 | 33 | 23.71 | 0.215 | 177 | 1.01E+01 | 0.103 | 223 |

**Acknowledgements**

The authors would like to acknowledge the Swedish EPA (Naturvårdsverket), CRAICC (Cryosphere-atmosphere interactions in a changing Arctic climate) and eSTICC (eScience Tools for Investigating Climate Change at High Northern Latitudes) for the financial support. The aerosol and meteorological data for Barrow and Tiksi were

5  downloaded from the International Arctic Systems for Observing the Atmosphere (www.iasoa.org) consortium website. Anne Jefferson (NOAA) and Eija Asmi (Finnish Meteorological Institute) provided their valuable insights on the Barrow and Tiksi datasets, respectively.

The work at Villum Research Station, Station Nord was financially supported by the Danish Environmental Protection Agency with means from the MIKA/DANCEA funds for Environmental Support to the Arctic Region,

10  which is part of the Danish contribution to "Arctic Monitoring and Assessment Program" (AMAP) and to the Danish research project "Short lived Climate Forcers" (SLCF). This work was also supported by the Nordic Centre of Excellence Cryosphere-Atmosphere Interactions in a Changing Arctic Climate (CRAICC). The Villum Foundation is acknowledged for funding the construction of Villum Research Station, Station Nord.

We would also like to express our appreciation and gratitude for the work and effort of all the scientists and

15  engineers involved in setting up and maintaining the Arctic aerosol sites, as well as making the data available and providing useful documentation.





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





**List of Figures**

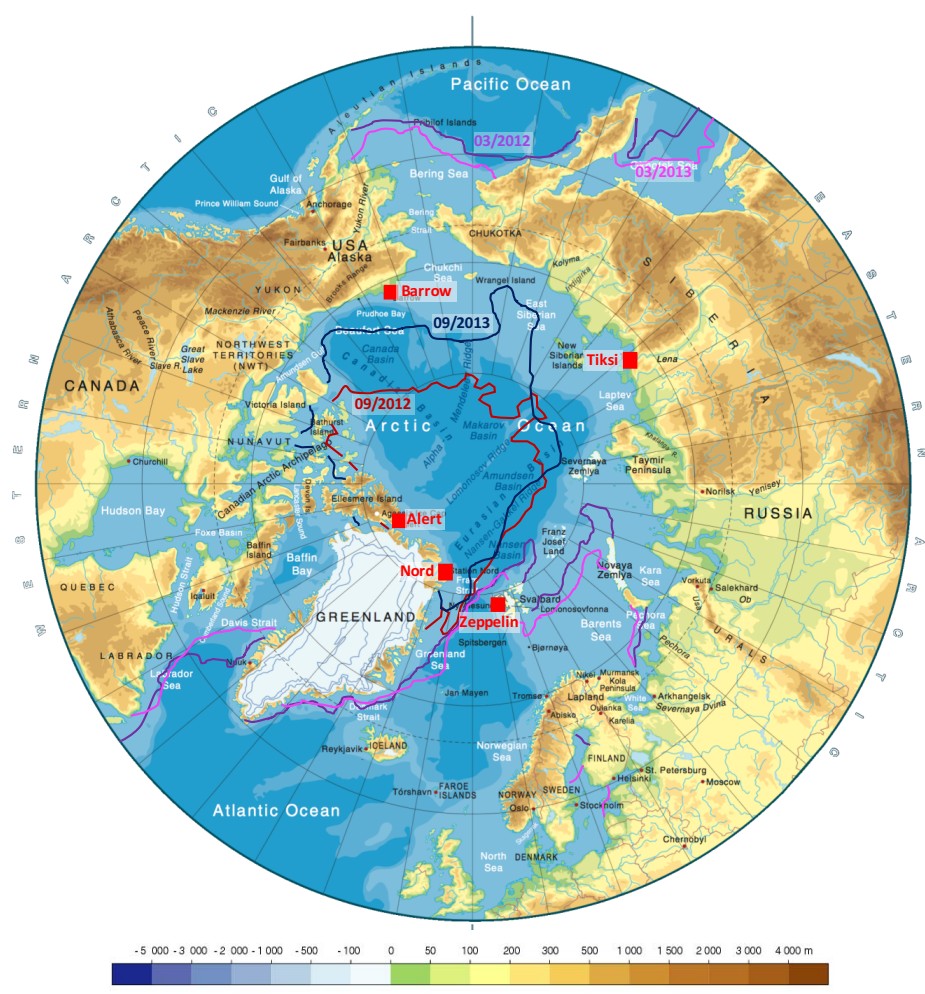

**Figure 1: A physical map of the Arctic region, with the five aerosol measurement sites marked with red squares around the Arctic ocean. The curved lines indicate the areal extent of the satellite-derived minimum and maximum ice edge in 2012 and 2013, which occurred in September and March, respectively. Here, the ice edge is defined as where the ice concentration is 50%.**





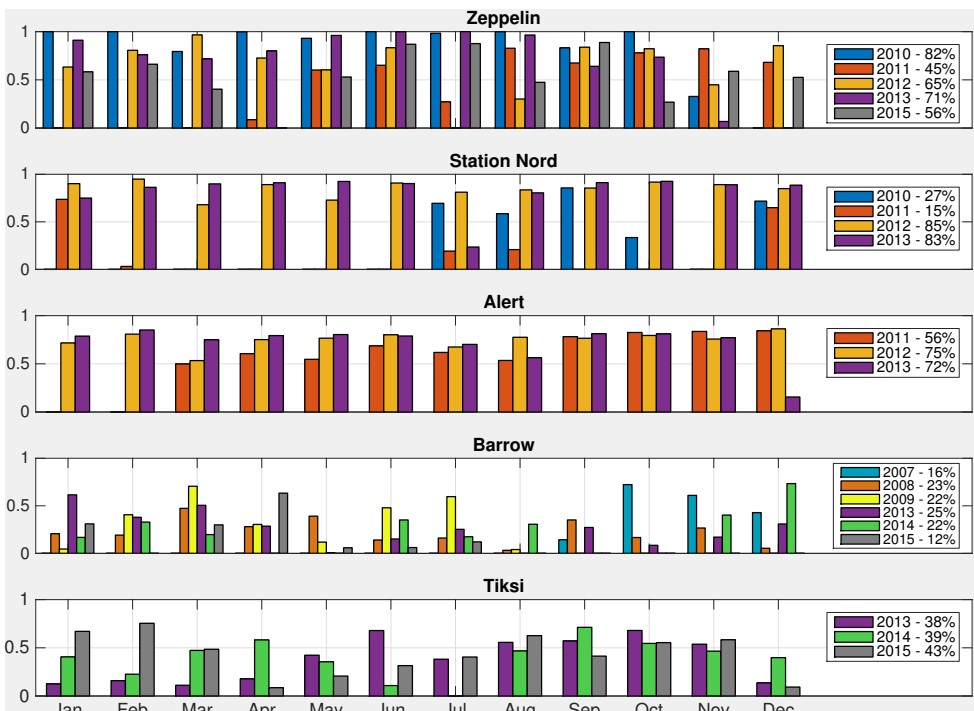

**Figure 2: Data availability.** The bars specify the fraction of the time within each month with available aerosol data after filtering (see Sect. 2.2). The colours of the bars represent different years, as shown in the legend, and the percentages therein indicates the yearly total data coverage.

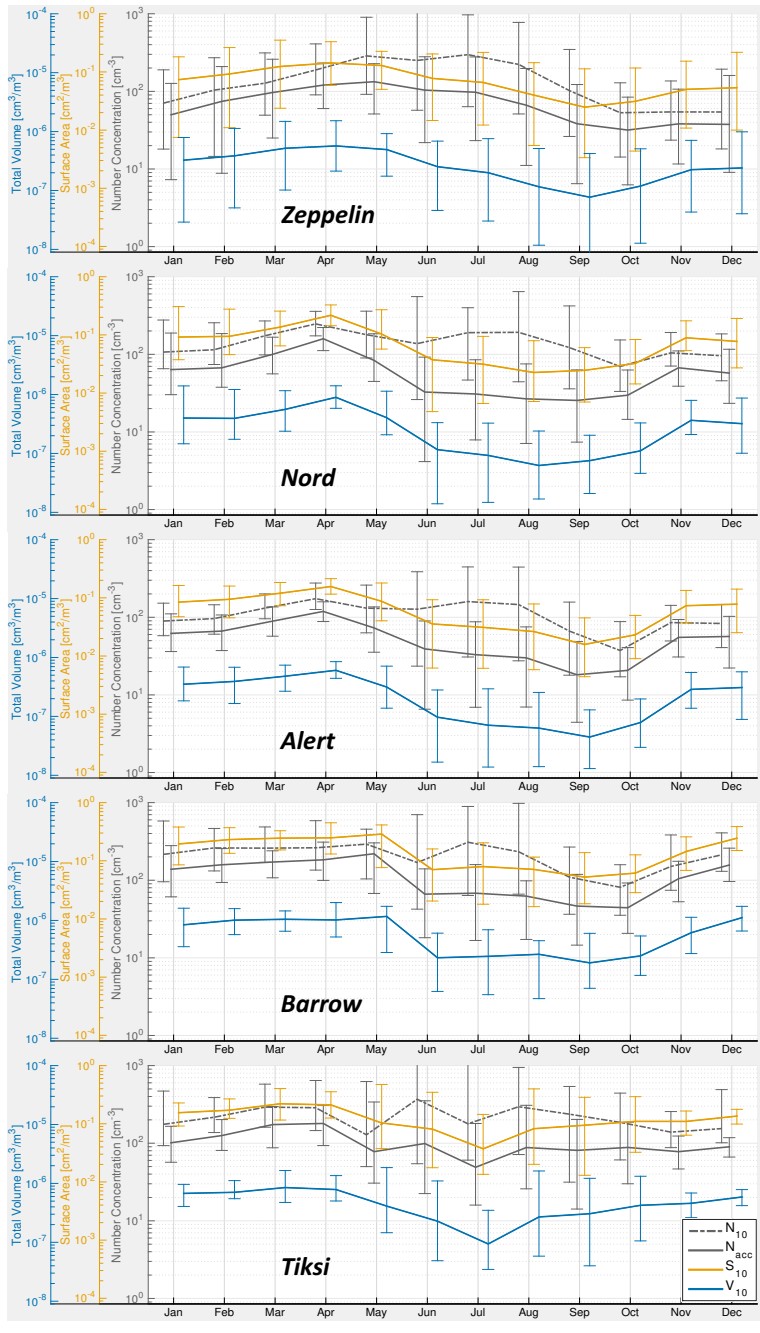

**Figure 3: Annual cycle of aerosol bulk properties.** The monthly median and interquartile ranges of the aerosol total and accumulation-mode number concentrations are indicated in grey dashed and solid curves and whiskers, respectively. The aerosol total surface area is in orange and the total volume in blue. The total values were calculated for the dry-diameter range of 10 to 500 nm, and the accumulation-mode size range is defined here between 100 and 500 nm. Particles were assumed spherical for the computation of the aerosol surface area and volume. The colour of the vertical axes and their labels, correspond to the colour of the curves, which are colour-blind friendly.



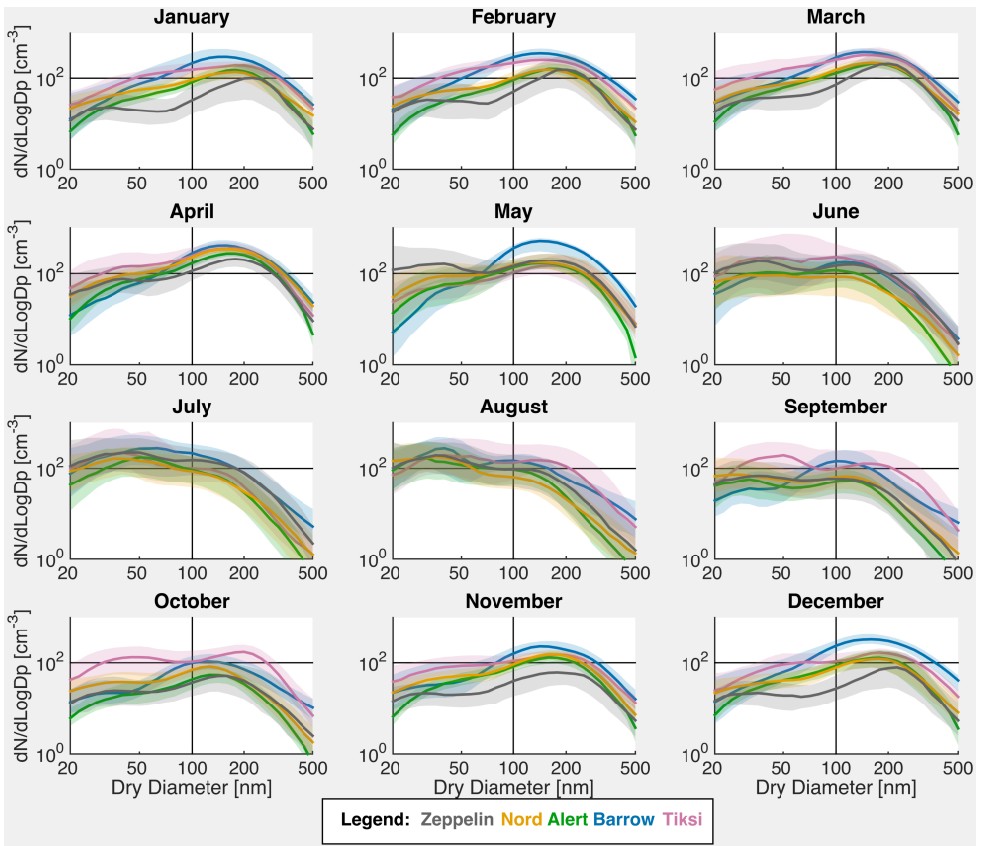

**Figure 4: The monthly aerosol number size distributions.** The solid curves indicate the median distributions of the 29-bin homogenized dataset, for each of the sites, with the colours matching the colours of the station name in the legend. The shaded semi-transparent areas denote the inter-quartile range and imply on the variability of the aerosol number size-distribution is each of the months. The horizontal and vertical black lines at 100 cm$^{-3}$ and 100 nm, respectively, are guidelines for facilitating the comparison between the months. The fitting parameters for approximating these aerosol spectra as the sum of three lognormal distributions, are given in Appendix A.




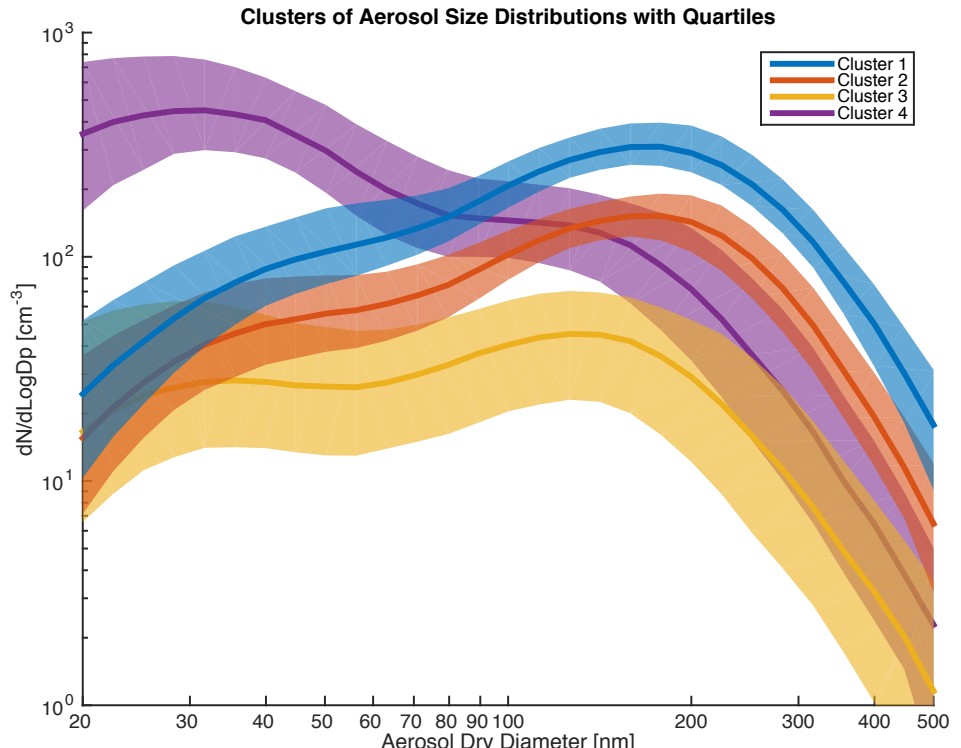

**Figure 5: The output of the K-means cluster analysis. The solid curves show the centroids (median) of the number size-distribution of the four clusters. The shaded area denotes the inter-quartile range of each of the size bins, for all distributions that were members of the given clusters.**

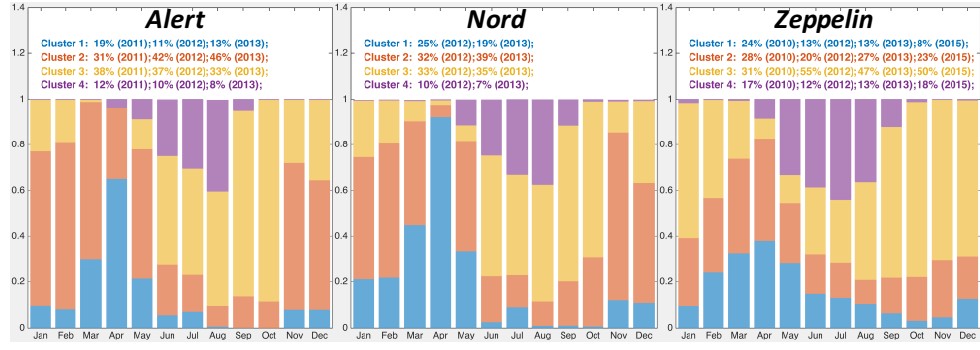

**Figure 6: The monthly probability of cluster occurrence. The bars indicate the mean relative frequency of the four aerosol clusters within each month (weighted by their number of occurrences in each of the included years). The colour coding is the same as in Fig. 5. The percentages in the legend indicate the mean annual relative occurrence of each of the clusters, only for years that had a fair representation of all seasons (cf. Fig 2). Tiksi and Barrow were not part of the cluster analysis due to their poorer data coverage (Fig 2) and less representative wind-directional sampling (cf. Sect. 2.2).**





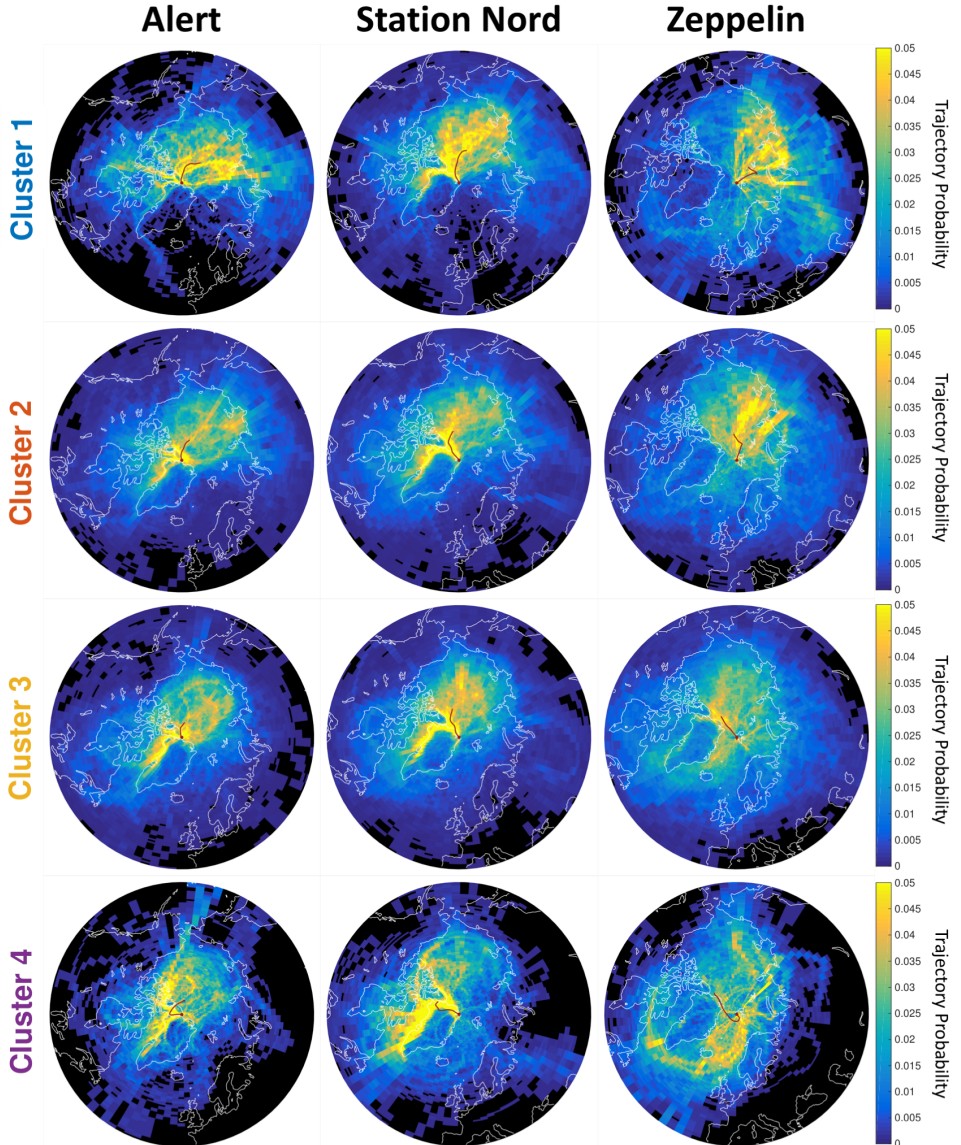

**Figure 7: The trajectory density associated with each aerosol cluster and site. The values represent the probability of the 240-hr long trajectories to cross the grid cells. Each cell is 0.5 and 4 degrees wide in latitude and longitude, respectively, in a concentric coordinate system whose pole is at the measurement site. The yellow shading indicates trajectory probability greater than 5%. The geographical mean location of the 240-hour long back-trajectories is indicated in red. They are much shorter than the individual trajectories because trajectories at one side of the site cancel out the trajectories on the opposite side in spatial averaging. Still, the mean location gives an indication of the direction from which most trajectories arrive to the site.**



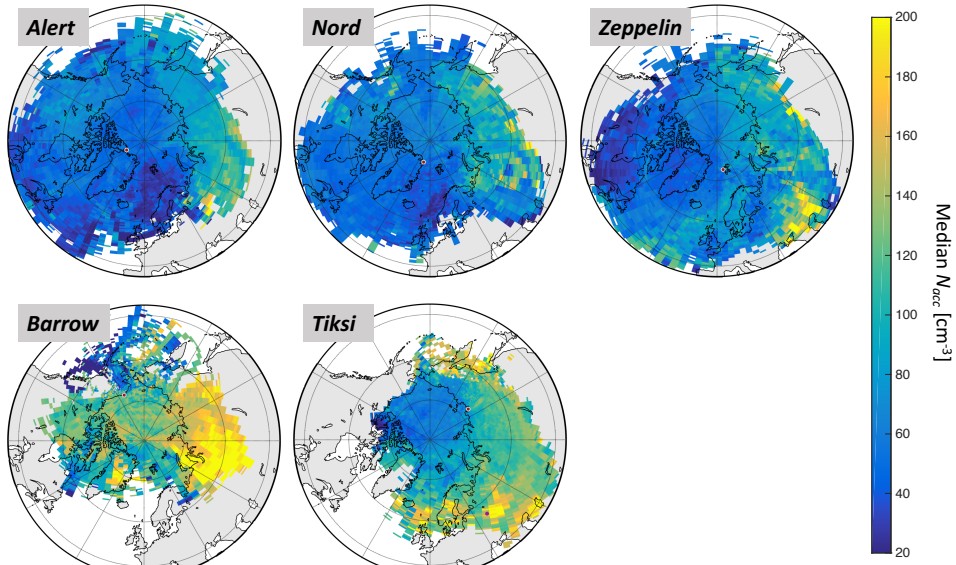

**Figure 8: Maps of median concentrations of accumulation-mode particles. Every 240-hr long back-trajectory is associated with the concentration that was measured at the site at the air mass arrival time – according to the trajectory analysis. The colour of each grid cell denotes the median concentration related to all trajectories crossing that cell. Each subplot represents a different site, whose location is indicated in red. All available data was included for each of the sites, but only grid cells with at least 10 trajectory passes are presented for statistical robustness.**

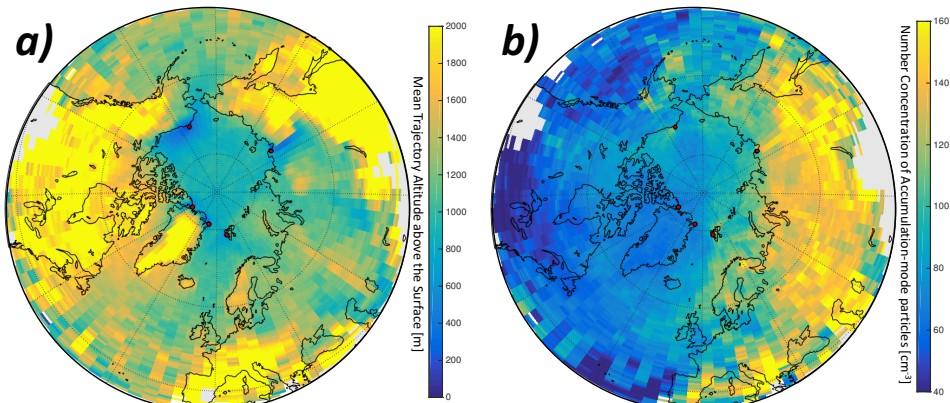

**Figure 9: Like Fig. 8, but the maps presents the combined the data for all sites together. Panel (a) displays the mean trajectory height above the surface, and panel (b) shows the mean concentration of accumulation mode particles. The locations of the sites are denoted by the small red circles.**





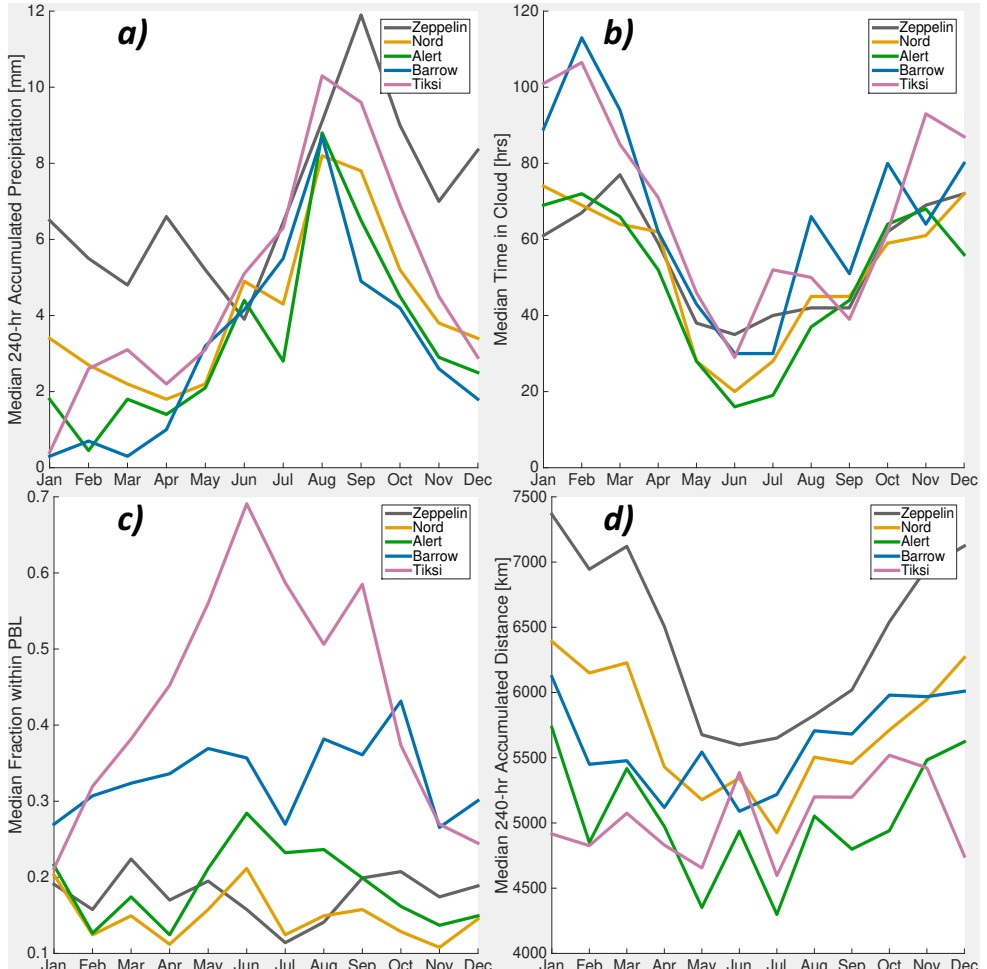

**Figure 10: Monthly statistics (median values) of trajectory-derived parameters for all sites. Panel (a): Accumulated precipitation along the 240-hr back-trajectory; Panel (b): Number of hours in a cloud. This was presumed when the relative humidity was greater than 95%; Panel (c): Fraction of trajectory time within the planetary boundary layer.**
5    **This was assumed when its altitude above the surface was smaller than twice the model mixing layer height; Panel (d): Trajectory accumulated distance.**





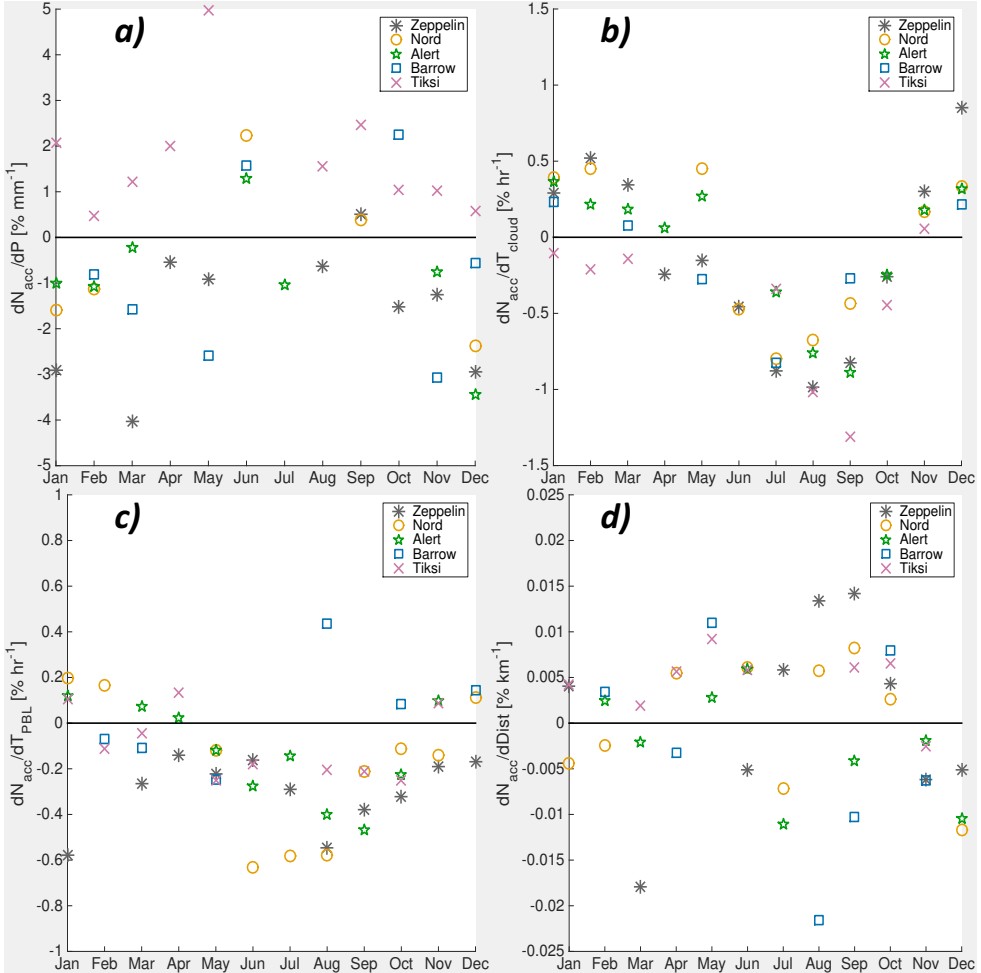

**Figure 11: The derivative of aerosol accumulation-mode concentration with each of the trajectory-derived parameters shown in Fig. 10 – for each month and site. Positive values indicate the percentage increase in accumulation-mode concentration with a unit increase in the corresponding parameter, while negative values show the opposite. Only statistically significant values (P<0.05) are shown.**





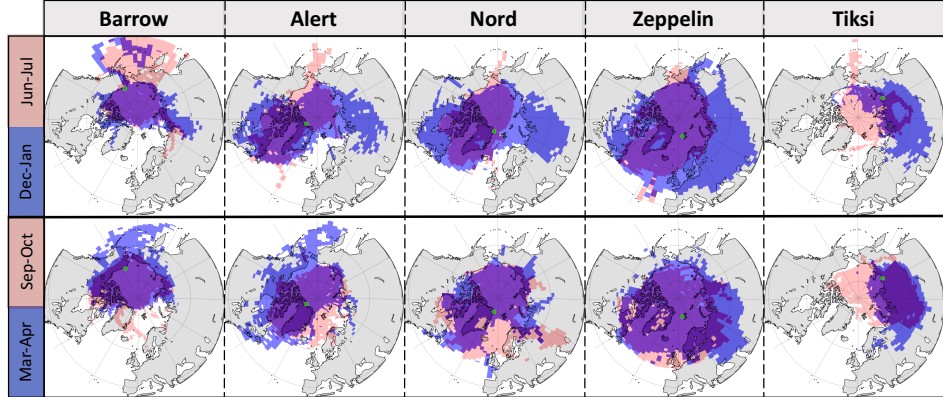

Figure 12: The seasonal areal coverage of 240-hr long back-trajectories for each of the sites. The top maps compare winter and summer in blue and red, respectively, while the bottom maps compare the spring and autumn. Only grid-cells with a trajectory-crossing probability greater than 0.5% are shown.





**List of Tables**

**Table 1: Quantitative information about the cluster centroids and members, which was the result of the k-means cluster analysis. The effective diameter, surface area, total volume and total mass are the centroid values, while the percentiles of the number concentrations are derived from all the cluster members.**

| | No. of Members | Effective Diameter [nm] | Surface Area [$cm^2\,m^{-3}$] | Total Volume [$cm^3\,m^{-3}$] | Total Mass [$\mu g\,m^{-3}$] | Number Conc. (10-500nm) [$cm^{-3}$] | | | Number Conc. (100-500nm) [$cm^{-3}$] | | |
|---|---|---|---|---|---|---|---|---|---|---|---|
| | | | | | | 25th | median | 75th | 25th | median | 75th |
| Cluster 1 | 5849 | 256 | 0.21 | $8.9\cdot10^{-7}$ | 1.34 | 165 | 206 | 290 | 125 | 150 | 204 |
| Cluster 2 | 12349 | 249 | 0.101 | $4.18\cdot10^{-7}$ | 0.63 | 79 | 102 | 145 | 59 | 72 | 99 |
| Cluster 3 | 12670 | 228 | 0.027 | $1.03\cdot10^{-7}$ | 0.15 | 23 | 36 | 78 | 13 | 19 | 40 |
| Cluster 4 | 3503 | 189 | 0.066 | $2.09\cdot10^{-7}$ | 0.31 | 170 | 273 | 464 | 39 | 56 | 97 |