# Peer review of "Pan-Arctic Aerosol Number Size Distributions: Seasonality and"

_Atmospheric Chemistry and Physics, 2017_

## Referee Comment (RC1) · Anonymous Referee #1 · 10 Feb 2017

This scientific paper gives an over view of the Number Size Distributions (NSD) measured at several monitoring sites located around the Arctic. Data is collected from 2010 to 2015 and using k-Means cluster analysis identifies 4 NSD - one nucleation and 3 accumulation. The paper then reviews the seasonality of these data and clusters - nucleation in the summer and accumulation (associated with Haze) in the darker spring-winter months. It also considers the sources giving rise to these NSD using the analysis of air mass back trajectories pointing to EuroAsia for the accumulation modes. The paper deals mainly with the accumlation mode particles and also shows that rain fall is important for aerosol scavenging. The paper gives a very useful observation of Arctic aerosol and recommended for publication.

Major Points None - well written paper.

Minor Points

Not wanting to interrupt the science, it would be useful to be able to refer to details of methodology in the Supplementary information. What type of normalisation was used before clustering and if one was not used how can a size dependence be avoided in the analysis?

The opening sentences of Section 3.3 are not completely clear. How was the decision to use 4 clusters arrived at? What happens if more clusters are chosen? Is there a spacial dependence when using 4 or more number of clusters? The output used to justify 4 clusters would be useful in the SI. Were validation statistics used and if so please present. Furthermore, was the whole 30,000 hour data set clustered at once or was there a limit set by the memory of the computer?

In figure 5, it would be interesting to see the minimum and maximum values plotted for each size bin to get a complete picture of the cluster output.

Pg 10 lines 35 onward, the explanation is not totally clear as to why the retreating ice and the trajectories in Figure 7 explain the skewed purple bars representing cluster 4 for Alert and Nord. Please clarify.

Pg 11 lin 18. For the mountain measurement site ZEP (looking beyond the widening of the Hoppel gap), how often does the boundary layer drop below the height of the station and what affect does this changing boundary layer have on the NSDs? And does a higher number of clusters reveal a different shape of NSD possibly associated with trajectories not associated with cluster 1 distributions?

Pg 13 line 40. How does the previous sentence referring to the height of the trajectories above 1km above the surface 'reinforce the claim that these areas are source region for accumulaton-mode particles in the Arctic? A detail or link is missing here - don't assume the reader will have prior knowledge.

Pg 14 lines 7-13. If they are of interest but you don't want to inflate the main text then

put such figures into the SI.

Appendix A should go in the SI because the values are not discussed or add to the value of the main paper.

---

## Referee Comment (RC2) · Anonymous Referee #2 · 23 Feb 2017

This is a scientifically sound, original and very well-written manuscript that should definitely be published in Atmospheric Chemistry and Physics. I have a few, mostly minor suggestions for clarifications when revising the paper.

Page 2, line 9: I would avoid using the term Anthropocene. Its starting point is ill-defined and it is very much debated how it should be defined.

Page 8, lines 31-36: I do not fully agree on the argumentation here. In a remote environment, why would an elevated site have more cloud-processed aerosols as a site close to the surface? For example, a marine boundary layer is usually well-mixed, so all the air in the BL is expected to circulated at about equal efficiency through clouds. When exposed to free-tropospheric air masses, air measured in Zeppelin is expected to be even less cloud-processed as air originating from the marine boundary layer,

as there tends to be less clouds in the free troposphere. I would rather think that are, in general, more clouds around Zeppelin than around other sites, causing more cloud-processing, or that clouds produce secondary material more efficiently around Zeppelin.

Page 9, end of section 3.2: I would like to see a few sentences here discussing the interpretation of the observed modal structure of the aerosol. When looking at the modal parameter in the tables in Appendix A, one can easily see that the recorded modes have some features that differ from the traditional classification of the submicron aerosol into the nucleation, Aitken and accumulation mode. First, there are always three modes, even though the nucleation mode is in most cases absent. Second, there are usually two modes in the size range usually called the accumulation mode. What are these modes? Condensation and droplet mode, as defined in some earlier studies, or something else. Third, the two largest mode are sometime very close to each other. Are they really separable when fitting the modes?

Page 16, lines 31-35: There is something wrong here. The text has a totally incomplete sentence that refers to a box model which is not explained anywhere and which does not seem to fit to the context of the surrounding text.

---

## Referee Comment (RC3) · Anonymous Referee #3 · 28 Feb 2017

The authors analyzed a multi-year observational dataset of aerosol number size distributions from five sites around the Arctic Ocean. They conducted cluster analysis and back-trajectory analysis to investigate the seasonality and transport patterns. This study could improve the understanding on the spatiotemporal variation and transport of air pollution over the Arctic region. Before this manuscript can be considered for publication, I have a few comments for the authors to address.

1. Page 2, Lines 30-31: For the authors' consideration, a very recent study (Qi et al., 2017) used an adjoint analysis to identify source regions of black carbon over the Arctic, which could be cited here.

References: Qi, L., Li, Q., Henze, D. K., Tseng, H.-L., and He, C.: Sources of Springtime Surface Black Carbon in the Arctic: An Adjoint Analysis, Atmos. Chem. Phys.

[Figure]

Discuss., doi:10.5194/acp-2016-1112, in review, 2017.

2. Measurement Section: the authors have provided qualitative descriptions on measurement sites and instrumentations. But more quantitative descriptions on the uncertainty associated with measurement instruments and methods as well as data processing should be provided in order to assess the validity of the analysis results in the text. Thus, I suggest including some quantitative discussions on the measurement uncertainties.

3. Page 4, Line 5: Please provide the full name of "DMA" here, since this is the first time when such abbreviation appears.

4. Page 6, Line13: "A 240-hour 3D back-trajectory ...". A more recent study (Qi et al., 2017, see comment #1 for reference) conducted adjoint model analyses to quantify source contributions of black carbon over the Arctic region and they found that the large contributions from Asian anthropogenic sources are mainly on 1-2 month timescales, which suggested that it is likely 5-day or 10-day trajectory analyses underestimate Asian contribution to surface BC in the Arctic. Could the authors add some discussions on this aspect, since a 10-day trajectory analysis is used in this study? Also for Line 17, Page 13, would the argument here still be valid if a longer time trajectory analysis is used?

5. Page 7, Line 16: "... comprising 29 bins ..." How much uncertainty could this re-scaling/re-distribution of size bins cause for the final analysis? Is there any specific reason for choosing 29 bins?

6. Section 3.3 (Cluster analysis): What is the accuracy of the k-means analysis to divide different clusters? Is there any way to quantify this? Would the different clusters also imply different aerosol components?

7. Figure 2: How would the data availability affect the final analysis? For example, for those years with a small fraction of available data at Tiksi and Barrow sites, would this

cause any bias in the analysis of seasonality and transport?

8. The HYSPLIT model analysis basically represents the transport of air pollution following the wind/flow (i.e., physical process). However, there are a lot of chemical productions of secondary aerosols during transport. How to deal with and interpret the source regions of these secondary aerosols formed in the middle of transport pathways? Would such secondary productions of aerosols affect the trajectory analysis?
* * *

---

## Author Comment (AC1) · 10 May 2017

This scientific paper gives an over view of the Number Size Distributions (NSD) measured at several monitoring sites located around the Arctic. Data is collected from 2010 to 2015 and using k-Means cluster analysis identifies 4 NSD - one nucleation and 3 accumulation. The paper then reviews the seasonality of these data and clusters - nucleation in the summer and accumulation (associated with Haze) in the darker spring-winter months. It also considers the sources giving rise to these NSD using the analysis of air mass back trajectories pointing to EuroAsia for the accumulation modes. The paper deals mainly with the accumulation mode particles and also shows that rain fall is important for aerosol scavenging. The paper gives a very useful observation of

[Figure]

Arctic aerosol and recommended for publication.

Major Points:

1. None - well written paper.

Minor Points:

1. Not wanting to interrupt the science, it would be useful to be able to refer to details of methodology in the Supplementary information. What type of normalisation was used before clustering and if one was not used how can a size dependence be avoided in the analysis?
Authors' Response: In sections 2.4 and 3.3 we describe the clustering methodology and provide relevant references. No "normalization" is done for this purpose so distributions with similar shapes but different concentrations, would be assigned a different cluster. However, we do homogenize the dataset, i.e. interpolate the number size distributions to a common number and bin-sizes (equally spaced on a logarithmic scale). The clustering algorithm itself doesn't "care" about the sizes of the particles but the same weight is given to every bin. If the bins would have been equally spaced on a linear scale, more (undesired) weight would have been given to the larger particles. We have clarified this in the revision.

2. The opening sentences of Section 3.3 are not completely clear. How was the decision to use 4 clusters arrived at? What happens if more clusters are chosen? Is there a spacial dependence when using 4 or more number of clusters? The output used to justify 4 clusters would be useful in the SI. Were validation statistics used and if so please present. Furthermore, was the whole 30,000 hour data set clustered at once or was there a limit set by the memory of the computer?

Authors' Response: We have revised the first paragraph in section 3.3 to justify the selection of four clusters as most suitable for the purpose of this paper as well as added a few references. The main points are that even if there are some objective ways to determine the "right" number of clusters, they may suggest a different number, and there is some room for subjective input.
There were no memory problems for running the cluster analysis on ~30,000 distributions.

3. In figure 5, it would be interesting to see the minimum and maximum values plotted for each size bin to get a complete picture of the cluster output.
Authors' Response: We chose to show the interquartile range, as showing a wider range (including minimum and maximum) would make the separation between the clusters very difficult and "overcrowd" the figure. The last paragraph in Sect. 3.3 discusses those "outliers".

4. Pg 10 lines 35 onward, the explanation is not totally clear as to why the retreating ice and the trajectories in Figure 7 explain the skewed purple bars representing cluster 4 for Alert and Nord. Please clarify.
Authors' Response: Page 11 lines 3-4 provides the explanation: high concentrations of DMS near the retreating ice edge (which continues to get closer to Alert and Nord even after the days get shorter and the sun is lower in the sky). We've added a reference to support that.

5. Pg 11 lin 18. For the mountain measurement site ZEP (looking beyond the widening of the Hoppel gap), how often does the boundary layer drop below the height of the station and what affect does this changing boundary layer have on the NSDs? And does a higher number of clusters reveal a different shape of NSD possibly associated with trajectories not associated with cluster 1 distributions?
Authors' Response: The determination of whether Zeppelin is within the mixed layer or not is not straight forward, due to the complex topography and often

stratified atmosphere with varying aerosol concentrations in the vertical dimension (we've added this and a reference to the manuscript). We only suggest this as another possible explanation for the reduced frequency of "Cluster 1" distributions at Zeppelin, compared to the Alert and Nord (along with increased North Atlantic effect, which may be more important). The slightly different median shapes of the NSD at Zeppelin might as well be due to its elevation, as stated in page 8 lines 28-36. However, increasing the number of clusters, as stated in the revised opening paragraph of section 3.3, mainly introduced additional clusters with differences in the concentrations of the smallest particles (i.e. splitting cluster 4 into sub-clusters). The effect of the added "exposure" to the free troposphere at Zeppelin is seemingly the increase in the frequency of higher cluster numbers (i.e. lower accumulation mode concentrations), rather than a significant change in the shape of the NSD.

6. Pg 13 line 40. How does the previous sentence referring to the height of the trajectories above 1km above the surface 'reinforce the claim that these areas are source region for accumulaton-mode particles in the Arctic? A detail or link is missing here - don't assume the reader will have prior knowledge.
   Authors' Response: The idea is that the air mass can "pick up" particles (and/or precursors) from sources at the surface, only if the trajectory height is close to the surface (e.g. within the PBL). Air masses passing high above the ground sources would not be affected by them. Therefore, to identify as source region for accumulation mode particles by this methodology, at least two conditions need to be met: 1) high mean/median Nacc concentration; 2) a low trajectory height. We rephrased and added some text in the manuscript for clarification.

7. Pg 14 lines 7-13. If they are of interest but you don't want to inflate the main text then put such figures into the SI.
   Authors' Response: Keeping the full data rather than plotting just the mean or median could provide additional information. However, in the case of the effect of

the extensive wild fires in Alaska in the beginning of July 2015, there is no need to provide a map of a higher percentile (or the maximum values), because they even show on the map of the median $N_{acc}$ for Zeppelin (Fig. 8) as well as the mean $N_{acc}$ map for all sites (Fig. 9b).

We've omitted the links in the footnote that refer to this event, and added a couple of recent references discussing the effects of this extreme event on aerosol observations in the European Arctic. We also rephrased the text to point to the effect of this event on the Zeppelin map.

8. Appendix A should go in the SI because the values are not discussed or add to the value of the main paper.

   Authors' Response: We chose to include the log-normal fitting parameters for the monthly median number distributions as an appendix, because they describe quantitatively the curves in Fig. 4 and are referred to from the main text (end of section 3.2). In addition, the numbers are more accessible to the readers this way so they can get a general impression of the values, without the need to download another file to their computer and open it in a different program (which they will probably do in case they'd like to actually use these fitting parameters in some way). The fitting parameters for the median number distributions, as well as for other percentiles, will be available in a digital format as supplementary material. See also point 3 in the response to Referee #2.

**Anonymous Referee #2**

This is a scientifically sound, original and very well-written manuscript that should definitely be published in Atmospheric Chemistry and Physics. I have a few, mostly minor suggestions for clarifications when revising the paper.

1. Page 2, line 9: I would avoid using the term Anthropocene. Its starting point is ill-defined and it is very much debated how it should be defined.

Authors' Response: OK. Changed that.

2. Page 8, lines 31-36: I do not fully agree on the argumentation here. In a remote environment, why would an elevated site have more cloud-processed aerosols as a site close to the surface? For example, a marine boundary layer is usually well-mixed, so all the air in the BL is expected to circulated at about equal efficiency through clouds. When exposed to free-tropospheric air masses, air measured in Zeppelin is expected to be even less cloud-processed as air originating from the marine boundary layer, as there tends to be less clouds in the free troposphere. I would rather think that are, in general, more clouds around Zeppelin than around other sites, causing more cloud-processing, or that clouds produce secondary material more efficiently around Zeppelin.

Authors' Response: This is an important point. But as the last sentence in this paragraph states it is not trivial to identify whether the observed aerosol sample is from an air mass that is coupled of de-coupled from the surface. The frequent stratification of the arctic lower troposphere, together with effects of the local topography and the surface properties, make the picture much more complex than e.g. subtropical marine boundary layer. In addition, Zeppelin is often in a cloud, but that doesn't necessarily mean that it is coupled with the surface layer. Anyway, the main probable cause for the difference between the sites, as you suggest, is the increased cloudiness around Zeppelin on a long term and regional scale. We have revised this paragraph in that light.

3. Page 9, end of section 3.2: I would like to see a few sentences here discussing the interpretation of the observed modal structure of the aerosol. When looking at the modal parameter in the tables in Appendix A, one can easily see that the recorded modes have some features that differ from the traditional classification of the submicron aerosol into the nucleation, Aitken and accumulation mode. First, there are always three modes, even though the nucleation mode is in most cases absent. Second, there are usually two modes in the size range

usually called the accumulation mode. What are these modes? Condensation and droplet mode, as defined in some earlier studies, or something else. Third, the two largest mode are sometime very close to each other. Are they really separable when fitting the modes?

Authors' Response: This is more of a technical thing. The fitting procedure takes user -defined inputs, such as the number of desired modes as well as ranges of acceptable values. These should be based on physical principles, otherwise the solution may converge to a "mathematical" solution, as there is more than one set of parameters that can fit the observations quite well (especially when it is a relatively narrow range that does not often have three distinct modes). The acceptable ranges for the fitting parameters that we originally used, were quite wide and single modes could overlap. The sum of the modes though still represented very well the observed number distributions, with the relative mean absolute error well below 0.2% with respect to the total number concentration for every fit. However, in order to place the single modes within the ranges of the nucleation, Aitken and accumulation-mode ranges, we have rerun this analysis and updated the tables (this did not affect the overall quality of the fit), as well as added a few clarifying sentences in this revision.

4. Page 16, lines 31-35: There is something wrong here. The text has a totally incomplete sentence that refers to a box model which is not explained anywhere and which does not seem to fit to the context of the surrounding text.

Authors' response: We've added a sentence to put this conceptual model in context and link it with the previous paragraph, as well as slightly modified the text and removed the term "box model".

**Anonymous Referee #3**

The authors analyzed a multi-year observational dataset of aerosol number size distributions from five sites around the Arctic Ocean. They conducted cluster analysis and back-trajectory analysis to investigate the seasonality and transport patterns. This study could improve the understanding on the spatiotemporal variation and transport of air pollution over the Arctic region. Before this manuscript can be considered for publication, I have a few comments for the authors to address.

1. Page 2, Lines 30-31: For the authors' consideration, a very recent study (Qi et al., 2017) used an adjoint analysis to identify source regions of black carbon over the Arctic, which could be cited here.
   References: Qi, L., Li, Q., Henze, D. K., Tseng, H.-L., and He, C.: Sources of Spring- time Surface Black Carbon in the Arctic: An Adjoint Analysis, Atmos. Chem. Phys. Discuss., doi:10.5194/acp-2016-1112, in review, 2017.
   Authors' Response: We added this citation.

2. Measurement Section: the authors have provided qualitative descriptions on measurement sites and instrumentations. But more quantitative descriptions on the uncertainty associated with measurement instruments and methods as well as data processing should be provided in order to assess the validity of the analysis results in the text. Thus, I suggest including some quantitative discussions on the measurement uncertainties.
   Authors' Response: The main point of this work, is to combine the observations from the individual sites and discuss them in the large perspective. We therefore include a rather short and mostly quantitative description of each site with the information that we find relevant. We also provide the references to the papers from the groups that were involved in making the measurements at the specific sites, where more details about the instrumentation, setup, calibrations and uncertainties can be found (although we have added in this revision the DMPS

uncertainties for Zeppelin).

What we have done was using the original datasets, and homogenzing them by interpolating the data to common bin sizes, range and temporal resolution (hourly). This part doesn't introduce any uncertainties, as we discuss features of a much longer temporal scales.

We've also filtered the data to remove observations that were potentially affected by local pollution and describe how this is done, and how that may affect our results (section 2.2). However, when a number is given, we provide different percentiles to demonstrate that range of variability, which is much larger that the uncertainty in the observations or analysis methods (e.g. Table 1, Figs. 3 and 4). We do provide an uncertainty estimate for the trajectory analysis (section 2.3), and also added in this revision the information about the relative error of the log-normal fitting results (appendix A).

3. Page 4, Line 5: Please provide the full name of "DMA" here, since this is the first time when such abbreviation appears.
   Authors' response: Done.

4. Page 6, Line13: "A 240-hour 3D back-trajectory ...". A more recent study (Qi et al., 2017, see comment #1 for reference) conducted adjoint model analyses to quantify source contributions of black carbon over the Arctic region and they found that the large contributions from Asian anthropogenic sources are mainly on 1-2 month timescales, which suggested that it is likely 5-day or 10-day trajectory analyses underestimate Asian contribution to surface BC in the Arctic. Could the authors add some discussions on this aspect, since a 10-day trajectory analysis is used in this study? Also for Line 17, Page 13, would the argument here still be valid if a longer time trajectory analysis is used?
   Authors' response: The reason for choosing 10 days is explained in the second paragraph of section 2.3 (balance between uncertainty in the trajectories and the typical lifetime of the particles). More than 10 days would not be suitable for the

kind of analysis and geographical resolution we use in this work.

In the discussion of the results, we do state that 10 days may not be enough to cover all possible sources when removal processes are not effective (last paragraph in section 3.5), and show that the sources are likely further away than the extent of the shaded area of the maps (page 13 last paragraph and Fig. 9). This is in line with the references provided, including with Qi et. al, 2017, which was added to the discussion.

The argument in Line 17 page 13 s only relevant, as stated, for a 10-day period. Extending this time frame would extend the areal coverage of the Nord, Alert and Barrow trajectories so they may cover the area between the Caspian and Aral seas, but the uncertainties of the trajectory locations around there would be greater/much greater than 1000 km, which is double the distance between the Aral and Caspian seas.

5. Page 7, Line 16: ". . . comprising 29 bins . . ." How much uncertainty could this re- scaling/re-distribution of size bins cause for the final analysis? Is there any specific reason for choosing 29 bins?

   Authors' Response: This doesn't add uncertainty as 29 bins can represent the hourly distributions (let alone the monthly distributions that are presented and analysed in the manuscript) very well and in detail.

   The range of the homogenized dataset is between 20 and 500†(actually 502) nm. Bin spacing of exactly 0.05 in the log space, yields 29 bins. Smaller spacing would not add any real information. We've added this information to the manuscript.

6. Section 3.3 (Cluster analysis): What is the accuracy of the k-means analysis to divide different clusters? Is there any way to quantify this? Would the different clusters also imply different aerosol components?

   Authors' response: The clustering of the aerosol number size distributions, using the k-means algorithm, has been done before and is very robust. By itself it

does not introduce any errors. There are different ways to evaluate how well the clusters represent the data, but the more relevant question is how many clusters to choose. Requesting more output clusters would inevitably reduce the total sum of errors and hence improve the accuracy, but at some point, only by little. There is no absolute "right" number of clusters to choose (even some different objective methods may suggest a different number for the same dataset), but we've evaluated the output using the "elbow" and "silhouette" methods. We have revised the first paragraph in section 3.3 (and added some references), where we try to justify our choice of four clusters.

Once we have done that, rerunning the analysis resulted in the very same output centroids (to confirm that there was no problem of the result converging to a local minimum). Of course, if different input is used (i.e. including more years etc.), the cluster means/medians may vary a little bit, but their general properties/features that are relevant for the discussion in our study would remain the same.

Different clusters may be related to different aerosol components, as they are linked with different source regions and seasons (Figs. 6 and 7). However, we only analysed the size distribution of the aerosols, and their chemical composition is not within the scope of this manuscript.

7. Figure 2: How would the data availability affect the final analysis? For example, for those years with a small fraction of available data at Tiksi and Barrow sites, would this cause any bias in the analysis of seasonality and transport?

   Authors' response: This depends on the specific analysis and how the data availability varies through the year. Generally, there is no problem if observations are missing for a part of a specific year - it is more important to have a decent representation for each of the months.

   For example, for the analysis of the relative monthly frequencies of the clusters (Fig. 6), the input data needs to be representative. Our requirement was to have at least two months (same month different years) with more than 50% data

coverage in each of them. Barrow and Tiksi does not fulfil this requirement and therefore were excluded from this specific analysis.

A potential issue with Tiksi and Barrow is the exclusion of the data from a specific (local) wind sector (the local wind direction may have some correlation with the large-scale wind flow). This may lead to a bias in the mean/median concentrations and reduce the representativeness of the remaining data (as stated in page 5 line 29). However, as discussed in the last paragraph in page 8, it doesn't seem to change the fact the both Tiksi and Barrow tend to have greater concentrations of accumulation-mode particles than at the other sites.

For the seasonality of the transport (Fig. 12) the right thing to do is to include all trajectories - even for the data that was suspected to be affected by local pollution. We have rerun this analysis for Barrow and Tiksi, updated Fig. 12 and added this information in section 3.5 (although the changes in the maps were very minor and did not affect the discussion).

8. The HYSPLIT model analysis basically represents the transport of air pollution following the wind/flow (i.e., physical process). However, there are a lot of chemical productions of secondary aerosols during transport. How to deal with and interpret the source regions of these secondary aerosols formed in the middle of transport pathways? Would such secondary productions of aerosols affect the trajectory analysis?

Authors' response: It is true that for obtaining the results shown in Fig. 8 (median $N_{acc}$ concentrations) the assumption is that no aerosol dynamics take place during the 10-day transport to the observation site. This requires an aerosol lifetime on that order or longer. This is why this analysis is not directly applicable of mapping the source regions of nucleation- or Aitken-mode particles (that lose their "identity" on shorter time scales). Also, part of the mass of the accumulation-mode particles observed at the Arctic sites, could originate from gaseous constituents (as stated in page 9 line 16), and be affected by cloud processing, deposition etc., which certainly take place during transport. This is why we don't claim that the median concentrations shown in Fig. 8, are directly comparable with real world values that would have been measured anywhere on the map, but rather refer to those concentrations "by association" and only use relative terms. It is definitely not an emission inventory map and where the particles have actually been emitted/formed could also be upwind or downwind of what seems to be a source region using this methodology. Without simulating the aerosol-related processes along the trajectories (and accounting for the mixing), it is very difficult to say how much the accumulation-mode particles changed in number/mass during the transport. This is highly variable, required different modelling tools and not within the scope of this paper. Even state-of-the-art models do not perform very well in simulating the aerosol concentrations and size distributions within the Arctic. This is why we decided not to assume anything about the aerosol dynamics of the accumulation mode particles (Figs. 8 and 9), but we do discuss in section 3.5 whether different processes would increase or decrease their concentrations during transport (and leave the more detailed analysis to a potential follow-up study (page 17 line 16)).
We've added a few sentences in section 3.4 to clarify this point.